# Fast Estimation of Partial Dependence Functions using Trees

**Jinyang Liu** [1]  **Tessa Steensgaard** [1]  **Marvin Wright** [2 3 4]  **Niklas Pfister** [1 5]  **Munir Hiabu** [1]

## Abstract

Many existing interpretation methods are based on Partial Dependence (PD) functions that, for a pre-trained machine learning model, capture how a subset of the features affects the predictions by averaging over the remaining features. Notable methods include Shapley additive explanations (SHAP) which computes feature contributions based on a game theoretical interpretation and PD plots (i.e., 1-dim PD functions) that capture average marginal main effects. Recent work has connected these approaches using a functional decomposition and argues that SHAP values can be misleading since they merge main and interaction effects into a single local effect. However, a major advantage of SHAP compared to other PD-based interpretations has been the availability of fast estimation techniques, such as `TreeSHAP`. In this paper, we propose a new tree-based estimator, `FastPD`*, which efficiently estimates arbitrary PD functions. We show that `FastPD` consistently estimates the desired population quantity – in contrast to path-dependent `TreeSHAP` which is inconsistent when features are correlated. For moderately deep trees, `FastPD` improves the complexity of existing methods from quadratic to linear in the number of observations. By estimating PD functions for arbitrary feature subsets, `FastPD` can be used to extract PD-based interpretations such as SHAP, PD plots and higher-order interaction effects.

---

[1]Department of Mathematical Sciences, University of Copenhagen, Denmark [2]Faculty of Mathematics and Computer Science, Universität Bremen, Germany [3]Leibniz Institute for Prevention Research and Epidemiology – BIPS, Germnay [4]Department of Public Health, University of Copenhagen, Denmark [5]Lakera, Switzerland. Correspondence to: Jinyang Liu <jl@math.ku.dk>.

*Proceedings of the $42^{nd}$ International Conference on Machine Learning*, Vancouver, Canada. PMLR 267, 2025. Copyright 2025 by the author(s).

*The implementation is available as an R-package on GitHub: https://github.com/PlantedML/glex

## 1. Introduction

As machine learning models become increasingly complex and widely deployed in mission-critical applications, interpretability has become essential for ensuring fairness and transparency (Adadi & Berrada, 2018). One popular model explanation method is Shapley additive explanations (SHAP) (Lundberg et al., 2020) – a post-hoc explanation method that has gained traction for its game-theoretic approach of attributing feature importance based on Shapley values. A value function must be specified for the Shapley value. In this paper we refer to SHAP as the Shapley values that use the partial dependence (PD) functions as the value function. Others (Chen et al., 2020; Taufiq et al., 2023) have termed it interventional Shapley values.

PD plots, i.e. one-dimensional PD functions (Friedman, 2001; Hastie et al., 2009; Molnar et al., 2023) quantify the effect of individual features by averaging predictions while holding a target feature at fixed values. However, as noted by Hiabu et al. (2023), both PD plots and SHAP do not provide a complete model interpretation. SHAP merges main effects and interactions while PD plots ignore interactions thereby limiting insight into feature contributions.

Alternatively, a functional decomposition allows for greater insight by clearly separating main effects and higher-order interactions. This idea has previously been investigated in Stone (1994); Hooker (2007); Chastaing et al. (2012); Lengerich et al. (2020); Herren & Hahn (2022) under the functional ANOVA identification constraint and more generally in Bordt & von Luxburg (2023); Fumagalli et al. (2025). In this paper, we consider an identification constraint based on PD functions for which we develop a fast algorithm.

Computational efficiency improvements for SHAP have been proposed in several contexts: model-agnostic methods such as FastSHAP (Jethani et al., 2022); deep-network adaptations (Ancona et al., 2019; Wang et al., 2022); and tree-based approaches via TreeSHAP (Lundberg et al., 2020). The popular path-dependent variant of TreeSHAP—used in XGBoost (Chen & Guestrin, 2016) and LightGBM (Ke et al., 2017)—is derived from approximating PD functions (Friedman, 2001). Recent work has further optimized these algorithms (Yang, 2022; Yu et al., 2022) and extended them to efficient SHAP interaction computations (Muschalik et al., 2024).

## 1.1. Contribution

We propose `FastPD`, a novel and efficient tree-based algorithm for consistently estimating arbitrary PD functions. `FastPD` can be used to obtain well-known PD-based explanations such as SHAP and PD plots. In addition it can be used to extract a functional decomposition that fully characterizes the target function with little computational cost. Finally, we show that path-dependent `TreeSHAP` can be an inconsistent estimate of the population SHAP value and demonstrate in a simulations study that there can be substantial differences between the path-dependent estimates and `FastPD` estimates. The remainder of this work is structured as follows. Section 2 introduces PD functions and their role in functional decomposition. Section 3 discusses estimation methods, computational complexity, and presents `FastPD`. Section 4 compares `FastPD` with existing PD-based explanation techniques.

## 1.2. Notation

For all $k \in \mathbb{N}$, we let $[k] := \{1, \ldots, k\}$, and for any subset $S \subseteq [d]$, define $\overline{S} := [d] \setminus S$. For $x \in \mathcal{X} \subseteq \mathbb{R}^d$, we use the notation $x_S \in \mathcal{X}^S$ to represent the coordinates of $x$ corresponding to the indices in $S$. Random variables are denoted by capital letters. Lastly for a $d$-dimensional function $m : \mathcal{X} \longrightarrow \mathbb{R}$, with a slight abuse of notation, we will write $m(x_S, x_{\overline{S}})$, nevertheless with the interpretation that the coordinates are permuted into the right order before applying $m$.

## 2. Motivation: PD-Based Explanations

Consider a real multivariate function $m : \mathbb{R}^d \supseteq \mathcal{X} \longrightarrow \mathbb{R}$ and a distribution $P_X$ on $\mathcal{X}$ with full support. For example, $m$ could be a black-box machine learning model for estimating the credit score of a customer and $P_X$ a distribution describing the customer base (see also Section 2.1). Our goal is to understand how changes to individual coordinates affect the function value. To this end, we consider a functional decomposition $\{m_S \mid S \subseteq [d]\}$ of $m$ such that for all $x \in \mathcal{X}$

$$m(x) = m_0 + \sum_{k=1}^{d} m_k(x_k)$$
$$+ \sum_{k<l} m_{kl}(x_{k,l}) + \cdots + m_{1,\ldots,d}(x)$$
$$= \sum_{S \subseteq [d]} m_S(x_S). \tag{1}$$

Unfortunately, without further assumptions such a decomposition is not unique. We will consider an identification strategy due to Hiabu et al. (2023), and assume that for all

$S \subseteq [d]$ and $x \in \mathcal{X}$

$$\sum_{U \subseteq S} m_U(x_U) = v_S(x_S), \tag{2}$$

where $v_S : \mathcal{X}^S \longrightarrow \mathbb{R}$ are the PD functions defined as

$$v_S(x_S) := \mathbb{E}_{P_X}[m(x_S, X_{\overline{S}})]. \tag{3}$$

The identification constraint (2) leads to the unique solution

$$m_S(x_S) = \sum_{U \subseteq S} (-1)^{|S \setminus U|} v_U(x_U), \tag{4}$$

which is known as the Möbius inverse (Rota, 1964) in combinatorics and as the Harsanyi dividend (Harsanyi, 1963) in cooperative game theory.

The PD function $v_S$ can be interpreted as the expected value of the function $m$ if the coordinates $x_k$ for all $k \in S$ are kept fixed while the remaining coordinates $k \in \overline{S}$ vary according to the distribution $P_{X_{\overline{S}}}$. This interpretation of PD functions directly carries over to sums of the components in the functional decomposition of the form $\sum_{U \subseteq S} m_U(x_U)$ via (2). Furthermore, the functional decomposition decomposes the function $m$ into additive contributions that together make up the whole function $m$. To illustrate this, consider the following two-dimensional example.

*Example* 2.1 (Functional decomposition). Assume that for $x_1, x_2 \in \mathbb{R}$ the PD functions take values

$$v_0 = 5, \ v_1(x_1) = 10, \ v_2(x_2) = 3, \ v_{1,2}(x_1, x_2) = 12.$$

Then from (4) the functional decomposition can be obtained as

$$m_0 = 5, \ m_1(x_1) = 5, \ m_2(x_2) = -2, \ m_{1,2}(x_1, x_2) = 4,$$

with the interpretation that fixing the $x_1$-value adds $m_1(x_1) = 5$ to the baseline prediction $v_0 = m_0 = 5$, leading to $v_1(x_1) = 10$. On the other hand, fixing both $x_1$ and $x_2$ and assuming no interaction, we would expect an output of $m_0 + m_1(x_1) + m_2(x_2) = 5 + 5 - 2 = 8$, but since the actual expectation is $v_{1,2}(x_1, x_2) = 12$, we have an interaction effect of $m_{1,2}(x_1, x_2) = 4$.

Two popular quantities used to capture feature contribution are PD plots and SHAP values. The latter can be defined for all $k \in [d]$ and all $x \in \mathcal{X}$ via a game theoretical motivation (Strumbelj & Kononenko, 2010) as

$$\Delta(S, k, x) := v_{S \cup \{k\}}(x_{S \cup \{k\}}) - v_S(x_S),$$

$$\phi_k(x) := \sum_{S \subseteq [d] \setminus \{k\}} \binom{d-1}{|S|}^{-1} \cdot \Delta(S, k, x). \tag{5}$$

Hiabu et al. (2023) showed that a functional decomposition can represent both the PD plots and the SHAP values as

$$v_k(x_k) = m_0 + m_k(x_k), \qquad \text{(PD plot)}$$

$$\phi_k(x) = m_k(x_k) + \frac{1}{2}\sum_j m_{kj}(x_{kj}) + \cdots$$

$$+ \frac{1}{d}m_{1,\ldots,d}(x_{1,\ldots,d}). \qquad \text{(SHAP value)}$$

These expansions illustrate that PD plots capture the main effects in the functional decomposition at a specific coordinate value $x_k$, while SHAP values aggregate main effects and interaction effects of all orders. As a result, neither PD plots nor SHAP values fully capture the behavior of $m$, leading to potential misinterpretations as illustrated in Example 2.2 – a similar argument is made by Hiabu et al. (2023).

*Example* 2.2 (PD plots and SHAP values do not fully capture $m$). Consider the function $m : \mathbb{R}^2 \longrightarrow \mathbb{R}$ defined for all $x \in \mathbb{R}^d$ by $m(x_1, x_2) \coloneqq x_1 + 2x_1x_2$, and let $P_X$ be a distribution with mean zero. Then, $\{m_0, m_1, m_2, m_{1,2}\}$ defined for all $x \in \mathbb{R}^2$ by

$$m_0 = 2\mathbb{E}_{P_X}[X_1X_2], \quad m_1(x_1) = x_1 - 2\mathbb{E}_{P_X}[X_1X_2],$$

$$m_2(x_2) = -2\mathbb{E}_{P_X}[X_1X_2],$$

$$m_{1,2}(x_1, x_2) = 2x_1x_2 + 2\mathbb{E}_{P_X}[X_1X_2],$$

is the unique functional decomposition satisfying (1) and (2). For the SHAP value $\phi_1$ we get

$$\phi_1(x_1, x_2) = x_1 + x_1x_2 - \mathbb{E}_{P_X}[X_1X_2].$$

The PD plot $m_0 + m_1$ only captures the average dependence on $x_1$, which does not provide any insight on the interaction between $x_1$ and $x_2$. Similarly, the SHAP value $\phi_1$ only captures part of $m$ as it down-weights the interaction contribution between $x_1$ and $x_2$.

Instead of focusing on PD plots and SHAP values, we therefore advocate to estimate the full functional decomposition $\{m_S \mid S \subseteq [d]\}$ defined via (1) and (2). However, this entails estimation of all PD functions $\{v_S \mid S \subseteq [d]\}$. Unlike for SHAP values, where many fast estimation techniques are available (e.g., `TreeSHAP`), no fast algorithms have been proposed to estimate all PD functions. In Section 3 we propose such an algorithm that efficiently estimates all PD functions which can then be further used to extract PD-based explanations such as SHAP values and the functional decomposition with no significant extra cost.

### 2.1. What Is The Target $m$?

It is worth differentiating two use-cases of PD-based explanations: (i) When the function $m$ represents a pre-trained black-box model $\hat{m}$, such as a neural network or tree ensemble, and the sole focus is to understand the model

without drawing conclusions about the underlying data-generating process. (ii) When the emphasis is on the relationship between input features $X$ and a response $Y$. In this scenario, the machine learning model $\hat{m}$ is used to approximate a target function $m^*$ (e.g., the conditional mean $m^* : x \mapsto \mathbb{E}[Y \mid X = x]$ or the conditional quantile $m^* : x \mapsto \inf\{t \in \mathbb{R} \mid \mathbb{P}(Y \leq t \mid X = x) \geq \alpha\}$) which is the actual function we wish to explain.

It has been argued (e.g. Chen et al., 2020) that for the case (ii), when $m^*$ is the conditional expectation function, it can make sense to consider the functions $x^S \mapsto \mathbb{E}_{P_X}[Y \mid X_S = x_s] = \mathbb{E}_{P_X}[m^*(x_s, X_{\overline{S}}) \mid X^S = x^s]$ instead of PD functions. Janzing et al. (2020) argue from a causal perspective that PD functions are generally preferable to this approach and easier to interpret. We tend to agree with their arguments and only consider PD functions as defined in (3).

In this paper, we consider both cases $m = \hat{m}$ and $m = m^*$ as possible targets. We formally distinguish them by considering PD functions in two settings:

(i) The model PD function

$$v_S^{\hat{m}}(x_S) = \mathbb{E}_{P_X}[\hat{m}(x_S, X_{\overline{S}})]. \qquad (6)$$

(ii) The ground truth PD function

$$v_S^{m^*}(x_S) = \mathbb{E}_{P_X}[m^*(x_S, X_{\overline{S}})]. \qquad (7)$$

In both cases, PD-explanations are applied to a trained machine learning model $\hat{m}$. However, if the target is the ground truth PD function (ii), then additional assumptions are required to ensure valid explanations. An important point is that the ground truth PD function is only identified in settings in which $P^X$ is a product measure or regularity conditions are made that avoid unidentifiability due to extrapolation (e.g., assuming a parametric model). In contrast, such assumptions are not necessary in case (i), as it can, in fact, be of interest to understand the behaviour of $\hat{m}$ outside the training support.

Most algorithms for PD-based explanations rely on $\hat{m}$ being a specific machine learning model. This is also the case for `FastPD`, which supposes that $\hat{m}$ is a tree-based model. A possible work-around to this is to train a new (surrogate) tree-based model on $(X^{(1)}, \hat{m}(X^{(1)})), \ldots, (X^{(n)}, \hat{m}(X^{(n)}))$ and afterwards apply `FastPD` to this model. However, the additional approximation steps then lead to the same potential difficulties as in case (ii).

## 3. Estimation of PD Functions

The most direct approach to computing PD-based explanations is to directly compute the PD functions $v_S$, defined

*Table 1.* Comparison of the algorithmic complexity to estimate either the SHAP values or PD functions for all features and $n_e$ evaluation samples in a single decision tree using $n_b$ background samples. Here $d$ denotes the total number of features, $D$ the depth of the tree, $F$ the number of features the tree splits on and $R$ the operations required for a single model evaluation (for a single tree this is $O(D)$). The method of Friedman (2001) can estimate SHAP values from the PD functions (5).

| Method | Complexity (SHAP) | $v_S(x)$ ? | Details |
|---|---|---|---|
| `VanillaPD` | $O(R2^d n_e n_b)$ | Yes | Slow if $d$ is large, but applicable to all models |
| (Friedman, 2001) | $O(2^d 2^D n_e)$ | Yes | Only approximates PD functions, with same inconsistency as `TreeSHAP-path` |
| `TreeSHAP-path` | $O(D^2 2^D n_e)$ | No | Fast but inconsistent if features are correlated |
| `TreeSHAP-int` | $O(D 2^D n_e n_b)$ | No | Estimates are based on $\leq 100$ background samples[†] |
| (Zern et al., 2023) | $O(2^D n_b + 3^D D n_e)$ | No | Fast and consistent with any number of background samples |
| `FastPD` | $O(2^{D+F}(n_e + n_b))$ | Yes | Fast and consistent for any PD-based explanations |

in (3), for all $S \subseteq [d]$ required to compute the desired PD-based explanation. In most practical examples one does not have access to the data generating mechanism $P_X$ directly and therefore need to estimate it. For example, if $m$ is a prediction model for fraud detection and $P_X$ is the future distribution of customers' features the algorithm will be applied on, one may use (parts) of the current customer base as background data to approximate $P_X$. Here, we consider the case in which we observe a background sample $\mathcal{D}_{n_b} = \{X^{(1)}, \ldots, X^{(n_b)}\}$ consisting of $n_b$ iid samples from $P_X$ based on which we want to estimate the PD-based explanations. An obvious estimator in this setting is the empirical PD function, defined for all $S \subseteq [d]$ and all $x \in \mathcal{X}$ by

$$\hat{v}_S(x_S) = \frac{1}{n_b} \sum_{i=1}^{n_b} m(x_S, X_{\overline{S}}^{(i)}). \tag{8}$$

In a model-agnostic setting, using (8) to estimate the PD function $v_S$ for a fixed $S$ at a single evaluation point $x$ has a complexity of $O(Rn_b)$ where $R$ is the number of operations needed to evaluate $m$ at a single point. We usually refrain from evaluating the PD functions at all points in the domain $\mathcal{X}$, and instead only evaluate it at a set of $n_e$ evaluation points that depend on the precise application. Consequently, computing (8) for all $n_e$ evaluation points and all sets $S \subseteq [d]$ results in a complexity of $O(R2^d n_e n_b)$, which in many applications is intractable. We call this baseline approach of estimating PD functions `VanillaPD`. In Table 1 we compare its complexity to tree-based alternatives which we discuss next.

### 3.1. Tree-Based Methods

If $m$ is a decision tree, the complexity of obtaining various PD-based explanations can be substantially reduced. The

---

[†]See GitHub issue: https://github.com/shap/shap/issues/3461

earliest example we are aware of is an algorithm proposed by Friedman (2001) which proposed to compute PD functions by traversing the tree weighting predictions based on the coverage of each node. Assuming that $2^D < Rn_b$, where $D$ denotes the depth of the tree, it has a reduced complexity of $O(2^d 2^D n_e)$, compared to `VanillaPD`. The tree-based algorithm of Friedman (2001) is one of the only implementations we found that attempts to estimate the PD functions, however, by construction it only roughly approximates (8). As with `TreeSHAP-path` (discussed next) this can lead to inconsistent estimates (see Proposition 3.1). More Recent approaches have focused on estimating SHAP directly and hence do not provide estimates of the PD functions. The most notable algorithm is TreeSHAP (Lundberg et al., 2020), which comes in two variants: Path-dependent Tree-SHAP (`TreeSHAP-path`) and interventional TreeSHAP (`TreeSHAP-int`). We first examine `TreeSHAP-path`: By exploiting the tree structure, `TreeSHAP-path` reduces the factor $2^d n_b$ in the complexity of `VanillaPD` to $2^D D$. While this method is computationally efficient, the SHAP values it computes rely on the same approximation of the PD functions as the algorithm proposed by Friedman (2001). This leads to the undesirable property that for two distinct trees that have the exact same predictions, the estimated SHAP values may differ when the features are not independent. In particular for post-hoc explanations of a black box model, this dependence on the internals of the model is concerning. Moreover, even in the limit of infinite (background) data the SHAP values estimated by `TreeSHAP-path` do not necessarily converge to the model SHAP value. A formal statement of this inconsistency is provided in the following Proposition 3.1. A proof is given in the Appendix.

**Proposition 3.1** (Inconsistency of `TreeSHAP-path`)**.** *There exists a distribution $P_X$ on $\mathcal{X}$, evaluation point $x' \in \mathcal{X}$ and distinct trees $\hat{m}^A$ and $\hat{m}^B$ such that*

*(i)* $\forall x \in \mathcal{X} : \hat{m}^A(x) = \hat{m}^B(x)$, *but* $\hat{\phi}_k^{\hat{m}^A}(x', \mathcal{D}_{n_b}) \neq \hat{\phi}_k^{\hat{m}^B}(x', \mathcal{D}_{n_b})$,

*(ii)* $\lim_{n_b \to \infty} \left| \hat{\phi}_k^{\hat{m}^A}(x', \mathcal{D}_{n_b}) - \phi_k^{\hat{m}^A}(x') \right| > 0$     *a.s.,*

*where* $\mathcal{D}_{n_b} := \{X^{(i)}, \dots, X^{(n_b)}\}$ *consists of iid samples from* $P_X$, $\phi_k^{\hat{m}}(x')$ *denotes the population SHAP value computed via the model PD function and* $\hat{\phi}_k^{\hat{m}}(x', \mathcal{D}_{n_b})$ *denotes the* TreeSHAP-path *explanation of feature* $k$ *at* $x'$, *where* $\mathcal{D}_{n_b}$ *is used to compute the coverage probability of* $\hat{m}$.

TreeSHAP-int was proposed as a method that consistently estimates the SHAP values. However, it has a rather high complexity of $O(D2^D n_e n_b)$ that scales with the product $n_e n_b$. Recently, Zern et al. (2023) have reduced the complexity by considering all background samples simultaneously when traversing the tree, yielding an improved complexity of $O(2^D n_b + 3^D D n_e)$. The algorithm by Zern et al. (2023) is an improvement on TreeSHAP-int which however does not estimate PD functions and instead only estimates the differences $\Delta(S, k, x)$ in (5). While this approach reduces the complexity of estimating SHAP values it does not allow us to efficiently extract estimates of the PD functions.

In the following section, we propose FastPD which estimates all PD functions at a similar complexity to Zern et al. (2023). From this, the SHAP values and the complete functional decomposition $\{m_S \mid S \subseteq [d]\}$ can be extracted with practically no additional cost, providing a more complete explanation of $m$.

### 3.2. **FastPD** Algorithm

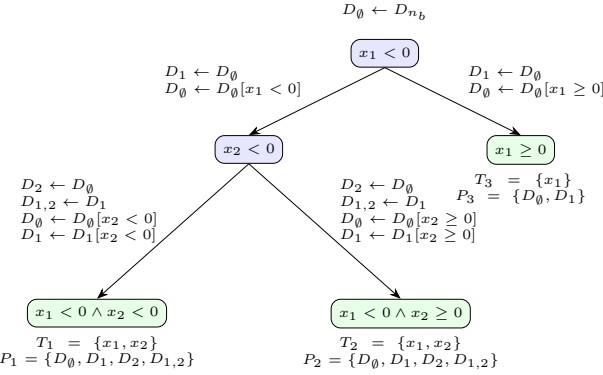

*Figure 1.* Augmentation step in FastPD. Augmentation entails calculating sets $D_S^{(j)}$ which contain observations that would have reached leaf $j$ if splits on features in $S$ were ignored. Starting from the full background sample $\mathcal{D}_{n_b}$ and setting $D_\emptyset \leftarrow \mathcal{D}_{n_b}$, each subsequent split creates new sets and updates existing ones. Here, $D_S[x_j < c] := \{i \in D_S : X_j^{(i)} < c\}$.

In this section, we introduce FastPD, which substantially reduces the complexity of computing $\hat{v}_S(x_S)$ for an evaluation point $x$ compared with VanillaPD. It operates in two main steps: (1) a tree augmentation step that precomputes partial dependence information using the $n_b$ background samples in $\mathcal{D}_{n_b}$; and (2) an evaluation step that retrieves partial dependence functions for any desired feature subsets on $n_e$ evaluation points.

The naïve estimator VanillaPD performs step 1 and step 2 for each evaluation point and background sample together, which leads to a complexity of $O(n_e n_b)$ in the number of the samples. The main observation used by FastPD is that the two steps can be separated entirely by exploiting the tree structure, thus resulting in a complexity of $O(n_e + n_b)$. To motivate the FastPD algorithm, we begin by noting that a decision-tree $\hat{m}$ can be expressed as a weighted sum of indicator functions. More concretely, there exists $L \in \mathbb{N}$, $c_1, \dots, c_L \in \mathbb{R}$ and $A_1, \dots, A_L \subseteq \mathbb{R}^d$ such that for all $x \in \mathcal{X}$

$$\hat{m}(x) = \sum_{j=1}^{L} c_j \mathbb{1}_{A_j}(x).$$

Furthermore, the leaves $A_j$ are rectangles, such that for each $j \in [L]$, $A_j := [a_{j1}, b_{j1}] \times [a_{j2}, b_{j2}] \times \cdots \times [a_{jd}, b_{jd}]$ determines the bounds of a leaf node in the tree. For all $S \subseteq [d]$ and $j \in [L]$ define the bounds of features $S$ on leaf $j$ as $A_j(S) := \prod_{s \in S} [a_{js}, b_{js}]$. By substituting $m$ with $\hat{m}$ in (8) and simplifying, we obtain

$$\hat{v}_S(x_S) = \sum_{j=1}^{L} c_j \underbrace{\mathbb{1}_{A_j(S)}(x_S)}_{(i)} \cdot \underbrace{\hat{P}\left(X_{\overline{S}} \in A_j(\overline{S})\right)}_{(ii)}, \quad (9)$$

where $\hat{P}$ denotes the empirical distribution of the background data $\mathcal{D}_{n_b}$, that is, $\hat{P}\left(X_{\overline{S}} \in A_j(\overline{S})\right) = n_b^{-1} \sum_{i=1}^{n_b} \mathbb{1}(X_{\overline{S}}^{(i)} \in A_j(\overline{S}))$. The factor (i) in (9) identifies the leaves in which $x$ would have landed if the splits corresponding to $\overline{S}$ were ignored during traversal, while the the factor (ii) in (9) represents the proportion of observations for which the features in $\overline{S}$ fall within the bounds of the leaf.

A naïve computation of (ii) in (9) loops over all observations and checks whether the feature in $\overline{S}$ lie within $A_j(\overline{S})$. The idea of FastPD is to separate the computation of partial dependence information on the background samples from evaluating on evaluation points into two phases:

**Augmentation** (Algorithm 1): For each leaf $j \in [L]$, we identify the set of features $T_j$ encountered along the path to that leaf. For every subset $S \subseteq T_j$, we save a corresponding list $D_S^{(j)}$ that contains the observations that would have reached leaf $j$ if splits on features in $S$ were ignored. For any sample $i$ it follows that $i \in D_S^{(j)}$ if and only if $X_{\overline{S}}^{(i)} \in$

$A_j(\overline{S})$. Consequently, factor (ii) in (9) can be computed efficiently as the ratio $|D_S^{(j)}|/n_b$.

**Evaluation** (Algorithm 2, Appendix B): Once the tree is augmented, the PD function $\hat{v}_S(x_S)$ at any fixed evaluation point $x$ and set $S$ can be computed quickly. Due to the augmentation step, every subset $S \subseteq T_j$ has a corresponding list $D_S^{(j)}$ saved in $P_j$ on leaf $j$. To compute $\hat{v}_S(x_S)$ for a point $x$ and features $S$, we can first intersect $S$ with the tree's split features to obtain a subset $U$ of features that is actively used by the tree. Afterwards, we traverse the tree to every leaf $j$ in which $x_U \in A_j(U)$.

The procedure guarantees that we arrive at the leaves $j$ in which the indicator (i) in (9) is equal to one. Once we are at leaf $j$, we can compute the empirical probability (ii) in (9) by counting the samples in $D_{U_j}^{(j)}$ where $U_j = U \cap T_j$. Note that $D_{U_j}^{(j)}$ exists in $P_j$ since $U_j \subseteq T_j$.

Lastly, because different subsets $S$ may lead to the same $U$ when intersected with the split features, redundant computations are avoided by saving $\hat{v}_U(x_U)$ (see lines 3 and 22 in Algorithm 2). For example, if the tree-depth is less than the number of features, then $U = \emptyset$ may occur often and the computed PD functions can be saved.

The consistency of `FastPD` follows from the consistency of the empirical estimator (8) which it computes exactly. A proof can be found in the Appendix.

### 3.2.1. COMPLEXITY OF FASTPD

The main reduction in complexity of `FastPD` stems from separating the computation of the partial dependence information on $D_{n_b}$ from the evaluation on $n_e$ evaluation points, which breaks a product into a sum.

To explicitly bound the algorithmic complexity of `FastPD` we can proceed as follows. Let $F$ be the number of unique features the tree has split. It will always hold that $F \le d$ and usually the inequality is strict. We start at the tree root with a list containing all observations and assign that to the set $S = \emptyset$, these lists are recursively passed to the child nodes. New lists $D_{S \cup \{k\}} := D_S$ are created for all $S$ when a new feature, $k$, is encountered on the path. This means that the total number of lists to keep track of will be at most $2^F$ on each node if every split feature was unique. For every node, every list incurs a maximum of $n_b$ operations, yielding a very rough bound of $O(2^{D+F} n_b)$ for the worst-case complexity of augmenting the whole tree. Once the tree has been augmented, the complexity of traversing all nodes is $O(2^D)$, and doing it for all $2^F$ subsets and evaluation points will result in a complexity of $O(2^{D+F} n_e)$.

Thus, if $D$ and therefore also $F$, are not too big – which usually is the case for gradient boosted trees as in as XGBoost

**Algorithm 1** `FastPD` augmentation step. Nodes are indexed by $j$, where $l_j$ and $r_j$ represent the indices of the left and right child nodes, respectively. The feature used to split at node $j$ is denoted by $d_j$, and $t_j$ is the split threshold and $v_j$ the value.

---

**input** Tree structure $= \{v, l, r, t, d\}$ and dataset $D_{n_b}$
**output** Augmented data $T = \{T_j \mid j \text{ is leaf}\}, P = \{P_j \mid j \text{ is leaf}\}$
1: **function** RECURSE $(j, T, P)$
2:  **if** $j$ is leaf **then**
3:    $T_j \leftarrow T$
4:    $P_j \leftarrow P$
5:    **return**
6:  **end if**
7:  **for all** $(S, D_S) \in P$ **do**
8:    **if** $d_j \in S$ **then**
9:      $P_{\text{yes}} \leftarrow P_{\text{yes}} \cup \{(S, D_S)\}$
10:      $P_{\text{no}} \leftarrow P_{\text{no}} \cup \{(S, D_S)\}$
11:    **else**
12:      $P_{\text{yes}} \leftarrow P_{\text{yes}} \cup \{(S, D_S[x_{d_j} < t_j])\}$
13:      $P_{\text{no}} \leftarrow P_{\text{no}} \cup \{(S, D_S[x_{d_j} \ge t_j])\}$
14:    **end if**
15:  **end for**
16:  $T_{\text{new}} \leftarrow T$
17:  **if** $d_j \notin T$ **then**
18:    $T_{\text{new}} \leftarrow T \cup \{d_j\}$
19:    **for all** $(S, D_S) \in P$ **do**
20:      $P_{\text{yes}} \leftarrow P_{\text{yes}} \cup \{(S \cup \{d_j\}, D_S)\}$
21:      $P_{\text{no}} \leftarrow P_{\text{no}} \cup \{(S \cup \{d_j\}, D_S)\}$
22:    **end for**
23:  **end if**
24:  RECURSE$(l_j, T_{\text{new}}, P_{\text{yes}})$
25:  RECURSE$(r_j, T_{\text{new}}, P_{\text{no}})$
26: **end function**
27: Initialize for all leaf nodes $j$: $T_j \leftarrow \emptyset, P_j \leftarrow \emptyset$
    RECURSE$(0, T = \emptyset, P = \{(\emptyset, D_{n_b})\})$

---

(Chen & Guestrin, 2016) and LightGBM (Ke et al., 2017) – even with a large number of features, the complexity of computing the PD function for all subsets $S$ is reduced. This is because the PD functions are calculated separately for every tree and then summed together. Hence for trees with moderate depth $D$, the main complexity does not stem from the number of subsets $S \subseteq [d]$ for which $v_S$ needs to be estimated, but in the traversal of all background samples for every new point that needs to be explained. If $n = n_b = n_e$, then both `VanillaPD` and `TreeSHAP-int` will scale proportionally to $n^2$. We hence gain a significant speed-up by reusing the computed empirical probabilities at the leaves whenever new points are explained.

**Proposition 3.2** (Consistency of `FastPD`). *Let $m : \mathcal{X} \to \mathbb{R}$ be a bounded target function and $P_X$ a distribution*

on $\mathcal{X}$. Then, for a sequence of iid background samples $X^{(1)}, X^{(2)}, \ldots \sim P_X$, it holds for all $S \subseteq [d]$ that $\lim_{n_b \to \infty} \hat{v}_{S,n_b}^m(x_S) = v_S^m(x_S)$ a.s., where $\hat{v}_{S,n_b}^m$ is the estimate for $v_S^m$ from FastPD applied with background data $\mathcal{D}_{n_b} = \{X^{(1)}, \ldots, X^{(n_b)}\}$. Moreover, if $\hat{m}_{n_b}$ is a uniformly consistent estimate of $m$ trained on $\mathcal{D}_{n_b}$, i.e., $\lim_{n_b \to \infty} \sup_{x \in \mathcal{X}} |\hat{m}_{n_b}(x) - m(x)| = 0$, a.s. then

$$\lim_{n_b \to \infty} \hat{v}_{S,n_b}^{\hat{m}_{n_b}}(x_S) = v_S^m(x_S) \quad a.s..$$

## 4. Experiments

We consider a supervised learning setup where training data $\mathcal{D}_n^{\mathrm{train}} := \{(Y^{(i)}, X^{(i)})\}_{i \in [n]}$ are iid sampled from a distribution $P$ with correlated features $X^{(i)}$. For all experiments, an XGBoost estimator, $\hat{m}$, was trained on $\mathcal{D}_n^{\mathrm{train}}$ and subsequently explained using the same training samples as background data.

Specific simulation settings varied depending on the analysis. For the inconsistency and MSE analyses (Figures 2, 3 and 5), we used $d = 2$ covariates sampled from a bivariate Gaussian distribution with correlation 0.3 and variance of 1. XGBoost hyperparameters (nrounds $\in \{1, \ldots, 1000\}$, eta $\in [0.01, 0.3]$, max_depth $\in \{2, \ldots, 6\}$) were tuned via 5-fold cross-validation with 50 random search evaluations. For the runtime comparison (Figure 4), we used $d = 7$ covariates sampled from a multivariate Gaussian distribution (details in Appendix) and a single fixed XGBoost model configuration with 20 trees and a maximum depth of $D = 5$. All other hyperparameters were left as default, except for eta, which was drawn uniformly from $[0.01, 0.3]$.

All simulations were conducted on a dedicated compute cluster (2 Intel Xeon Gold 6230 @ 2.1 GHz CPUs, 192 GB RAM). Our implementation of the FastPD algorithm is available as an R package on GitHub[‡].

**Inconsistency of TreeSHAP-path** Figure 2 illustrates the SHAP explanations of $\hat{m}$ for $X_1$ in 500 observations of $(Y, X) \in \mathbb{R} \times \mathbb{R}^2$. We observe that TreeSHAP-path inconsistently estimates the model SHAP obtained via the model PD function. In contrast, FastPD, which estimates the PD function based on the same 500 samples used as the background data, is consistent and lies close to the model SHAP. In the setting of Figure 2, $X_1$ and $X_2$ have a correlation of 0.3 further correlations of 0, 0.1, and 0.7 are considered in the Appendix.

**Non-Decreasing MSE of TreeSHAP-path**

Figure 3 depicts the mean squared errors (MSEs) of the different methods when the model SHAP is taken as the target. We observe that the MSE of TreeSHAP-path

[‡]R implementation of the FastPD algorithm: https://github.com/PlantedML/glex

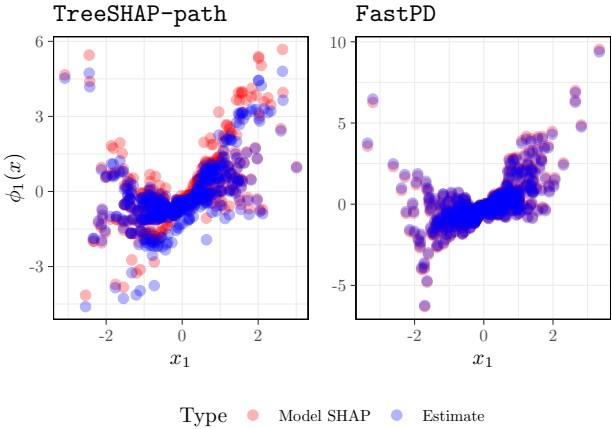

*Figure 2.* Comparison between FastPD (for SHAP) and TreeSHAP-path on simulated data of two covariates with correlation 0.3 and $n_b = 500$ background samples that are also used as evaluation points. For the two methods, the simulation run that achieved the median MSE was selected. We see that the model SHAP is captured well by the FastPD SHAP estimates (equivalent to VanillaPD) which is not the case for TreeSHAP-path.

does not shrink with increasing number of background samples, whereas the MSE of FastPD decreases substantially, demonstrating that it is consistent towards the model SHAP. Lastly, we observe that accurate estimation might require more than 100 background samples (The Python package shap e.g. does not use more than 100 background samples). In the setting of Figure 3 $X_1$ and $X_2$ have a correlation of 0.3 further correlations of 0 and 0.7 are considered in the Appendix. Notably even when the correlation is zero, FastPD turns out to be more accurate then TreeSHAP-path. This is because TreeSHAP-path does not leverage all information available in the background samples whereby at each inner node, it conditions on the path taken by the evaluation point.

**Comparison of Computational Runtime** Figure 4 compares the runtime of extracting the PD functions for all $S$ using FastPD with computing the interventional SHAP values as implemented in the SHAP Python package. An XGBoost model was pre-fitted with 20 trees and a max-depth of 5 on a fixed dataset of 8 000 observations. The number of background samples, $n_b$, was selected to be $1\,000, 2\,000, \ldots, 8\,000$. The same samples were used as evaluation points. VanillaPD and TreeSHAP-int scale quadratically in comparison to Zern et al. (2023) and FastPD, which has a linear complexity in the number of samples. While Zern et al. (2023) and FastPD have a comparable runtime, FastPD calculates all PD functions from which not only SHAP values but a complete functional decomposition can be derived.

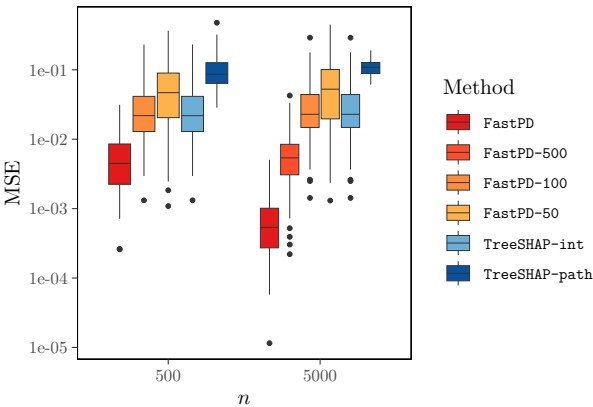

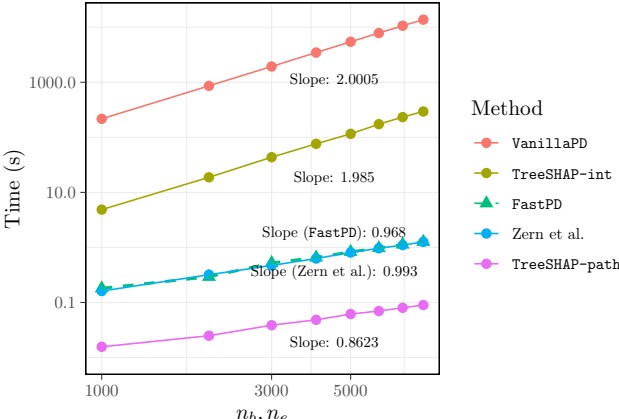

*Figure 3.* Comparsion of Mean Squared Error (MSE) of `TreeSHAP` versus `FastPD` over $B = 100$ simulations (log-scale). Each boxplot summarizes the results of $B = 100$ independent simulation runs. In each run, an XGBoost model $\hat{m}$ was trained using $n$ observations, and the same training samples were resampled as background data to estimate the partial dependence (PD) function. For `FastPD`, different numbers of background samples were used: $\{50, 100, 500\}$ for $n = 500$ and $\{50, 100, 500, 5000\}$ for $n = 5000$. SHAP values were estimated using the training observations as evaluation points, and the mean squared error (MSE) was computed as $\frac{1}{n}\sum_{i=1}^{n}(\phi_1(x^{(i)}) - \hat{\phi}_1(x^{(i)}))^2$, where $\phi_1$ denotes the model SHAP of feature $X_1$.

**Obtaining Functional Components** The functional components can be recovered from the estimated PD functions via (4). Figure 5 compares $m_1(x_1)$ from Example 2.2 with the functional component computed using `FastPD` and with the method proposed in Friedman (2001). The estimate of `FastPD` lies close to the component $\hat{m}_1(x_1)$ as one would have obtained via the model PD function, while the path-dependent (Friedman, 2001) suffers in areas outside the center. We also see that the component estimated by `FastPD-100` has a slope that is slightly off, which highlights the need to approximate the PD function with more background samples. In the setting of Figure 5, $X_1$ and $X_2$ have a correlation of $0.3$ and a setting with correlation of $0.7$ is given in the Appendix.

## 5. Real Data Experiments

### 5.1. Comparison on Benchmark Datasets OpenML-CTR23 and OpenML-CC18

We have conducted experiments on a significant number of curated datasets in both regression and classification settings. We used the OpenML-CTR23 (Fischer et al., 2023) regression datasets and the OpenML-CC18 (Bischl et al., 2021) classification datasets. In these experiments, we explained the predictions of a XGBoost model via functional decomposition as done in Figure 5. The hy-

*Figure 4.* Runtime in seconds (s) on log-log scale between `FastPD` and other methods as a function of the number of background samples and evaluation points which are taken to be the same. Measurements are the mean times over $B = 100$ runs where an XGBoost model was fitted on the data with 20 trees and a depth of 5 each. The mean times for $n_b = n_e = 1000$ are (217s, 4.86s, 0.181s, 0.161s, 0.0157s) respectively and (13722s, 295s, 1.26s, 1.26s, 0.0899s) for $n_b = n_e = 8000$. The runtime of `TreeSHAP-path` depends only on the number of evaluation points $n_e$.

perparameters `nrounds` $\in \{10, \ldots, 200\}$, `max_depth` $\in \{1, \ldots, 5\}$, `eta` $\in [0, 0.5]$, `colsample_bytree` $\in [0.5, 1]$ and `subsample` $\in [0.5, 1]$ were tuned via random search with 5-fold cross-validation over 100 random search evaluations. For each dataset, we computed the following measure of variable importance

$$\text{Importance}(\hat{m}_S) = \frac{1}{n}\sum_{i=1}^{n}\left|\hat{m}_S(X_S^{(i)})\right|. \quad (10)$$

Components $\hat{m}_S$ appearing in the top five of any method (`FastPD`, `FastPD-50`, `FastPD-100`, or `Friedman-path`) were compared in their importance attributions. Taking `FastPD` as the reference, the path-dependent algorithm exhibited relative differences exceeding $\pm 30\%$ in 33 of the 96 datasets. Applying `FastPD` with only 50 background samples (`FastPD-50`) reduced this to 18 datasets, and using 100 samples (`FastPD-100`) further lowered it to just 9. The full attribution results for each dataset are provided in the Appendix.

These experiments show that PD estimation accuracy improves with larger background samples. Our results suggest that a smaller background sample size may often suffice. Ideally, one would leverage the full dataset to compute PD functions; when this is not feasible, a practical strategy is to subsample (e.g. $n_b = 100$) and then checking whether the resulting PD functions closely match those from a larger sample (e.g. $n_b = 200$). Such a comparison provides a

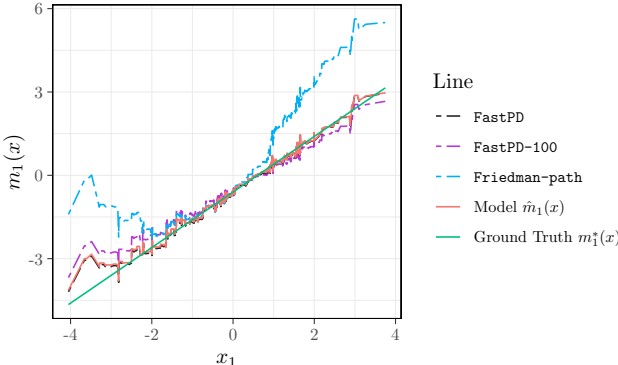

*Figure 5.* Comparison of estimated functional component $m_1$ between `FastPD` and the path-dependent method of (Friedman, 2001) (`Friedman-path`). The components are extracted from an XGBoost model trained on 5000 samples from the data-generating distribution. The model component ($\hat{m}_1$, red line) was computed by weighting the leaves using the true probabilities, while the ground truth component ($m^*$, green line) was computed analytically similar to Example 2.2.

simple yet effective check on the adequacy of the reduced background set.

### 5.2. The Adult Dataset

A particularly illustrative additional example is the `adult` dataset which contains data on whether an individual's income exceeds $50,000$ per year (Becker & Kohavi, 1996). We ran randomized grid search with 5-fold CV to tune XGBoost hyperparameters (`max_depth` $\in \{3,\dots,7\}$, `eta` $\in \{0.01, 0.05, 0.1\}$, `nrounds` $\in \{100, 200, 300\}$, `min_child_weight` $\in \{1, 3, 5\}$, `subsample` $\in \{0.6, 0.8, 1.0\}$, `colsample_bytree` $\in \{0.6, 0.8, 1.0\}$) over 25 trials. We then visualized the `age-relationship` interaction component (Figure 6) and noticed that `FastPD` and `Friedman-path` offered conflicting interpretations: With `FastPD`, we notice that at prime working age there is a slight positive effect on income for husbands, while the effect is close to zero and/or slightly negative for wives. In contrast, the path-dependent algorithm `Friedman-path` estimates the effect to be zero for both wives and husbands at working age; suggesting that age has the same effect on a husband's and wife's income in this range.

## 6. Discussion

In this paper, we have proposed `FastPD`, an algorithm that consistently estimates model PD functions for tree-based models $\hat{m}$ with linear time complexity in the number of observations. Under additional assumptions, it also consistently estimates the ground truth PD function of the condi-

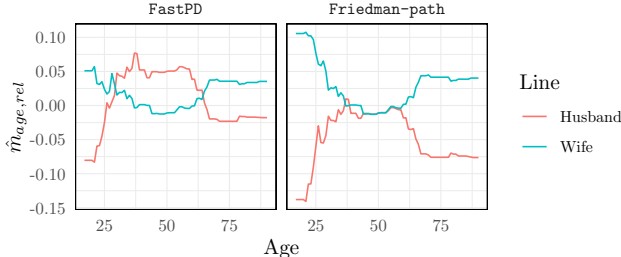

*Figure 6.* The component $\hat{m}_{age,rel}$ identified by `FastPD` and `Friedman-path` for the `adult` dataset. We see that `FastPD` estimates a positive effect for husbands during the prime working age, while `Friedman-path` estimates a zero effect for both husbands and wives.

tional expectation $m^*$. One limitation of Algorithm 2 is its space complexity, which increases with the number of lists at each leaf. However, once the tree is augmented, storing only the sample counts instead of the full lists can improve memory usage, but this problem also motivates limiting the tree depth. Fortunately, existing gradient boosting algorithms such as XGBoost perform well with shallow trees[§], as increasing depth is likely to cause the model to overfit.

It is important to distinguish the *model* PD function from the *ground-truth* PD function. If the target is the ground-truth PD, our recommendation is to use `FastPD` as an exploratory visualization tool, to be supplemented by semi-parametric or doubly-robust estimators (Kennedy et al., 2016; Chernozhukov et al., 2018).

Although `FastPD` can be used to recover functional components of any order, in practice higher-order interaction components may contribute negligibly; therefore one may choose to truncate the intitial black-box model at the main effects and pairwise interactions to improve interpretability. Additionally one can report the average discrepancy between the truncated model and the initial black-box model.

Finally, if the target is the ground-truth PD, a key challenge with PD-based explanations mentioned in Section 2.1 is that they can be distorted by extrapolation outside the support of $P_X$. Future work should investigate strategies to limit or correct for extrapolation, or consider alternative summaries such as average local effects $\mathbb{E}_{P_X}\left[\frac{\partial}{\partial x^j} m(X)\right]$ (ALE) (Apley & Zhu, 2020), which avoid these issues.

---

[§]XGBoost uses a default `max_depth` of 6, warning that deeper trees may increase memory usage aggressively: `https://xgboost.readthedocs.io/en/stable/parameter.html`

## Acknowledgments

Niklas Pfister was supported by a research grant (0069071) from Novo Nordisk Fonden. Marvin Wright is supported by the German Research Foundation (DFG) under the grants 437611051 and 459360854. Tessa Steensgaard and Munir Hiabu are supported by the project framework InterAct.

## Impact Statement

This paper contributes to the field of explainable AI (XAI) by providing a novel algorithm, `FastPD`, for estimating Partial Dependence (PD)-based explanations of tree-based models. XAI is a critical field focused on making complex machine learning models more transparent, trustworthy, and accountable. Our work addresses a key challenge in XAI: the need for computationally efficient methods that provide *consistent* and comprehensive explanations. While methods like SHAP are popular, some widely-used implementations can yield inconsistent results, potentially undermining trust. By enabling fast and consistent estimation of PD functions, `FastPD` allows for more reliable application of PD-based exaplanations, including SHAP and full functional decompositions. This enhanced reliability and the ability to disentangle main effects from interactions contribute directly to core XAI goals: fostering user trust, enabling robust model debugging, facilitating the identification and mitigation of bias for improved fairness, and supporting scientific discovery by providing deeper insights into model behavior.

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

# A. Additional Details on Simulations

In this section we provide additional details on the numerical experiments shown in Figures 2-5 in the main text. We first state the two data generating processes (DGPs) we used in the experiments. Afterwards, we will give details on the figures and specify which of the two DGPs has been used in each figure.

**DGP 1:** We consider the covariate distribution $P_X = \mathcal{N}(0, \Sigma)$ with covariance matrix $\Sigma = \begin{pmatrix} 1 & 0.3 \\ 0.3 & 1 \end{pmatrix}$ and the target function $m : \mathbb{R}^2 \longrightarrow \mathbb{R}$ is defined for all $x \in \mathbb{R}^2$ as

$$m(x) = x_1 + x_2 + 2x_1 x_2.$$

We generate independent samples from $(X, Y)$ by first sampling $X \sim P_X$ and then $Y \sim \mathcal{N}(m(X), 1)$.

**DGP 2:** We consider the covariate distribution $P_X = \mathcal{N}(0, \Sigma)$ with $\Sigma = 3 \cdot I_7 + \frac{3}{5} \cdot J_7$, where $I_7 \in \mathbb{R}^{7 \times 7}$ denotes the identity matrix and $J_7 \in \mathbb{R}^{7 \times 7}$ denotes the antidiagonal identity matrix (entries of ones going from lower left corner to upper right corner, rest being zero). The target function $m : \mathbb{R}^7 \longrightarrow \mathbb{R}$ is defined for all $x \in \mathbb{R}^7$ by

$$m(x) = 3 \sin(x_1) + 2.5 \cos(0.3 x_2) + 1.12 x_3 + \sin(x_4 x_5) + 0.7 x_6 x_7.$$

We generate independent samples from $(X, Y)$ by first sampling $X \sim P_X$ and then $Y \sim \mathcal{N}(m(X), 0.1)$. In Figure 3, this DGP is modified to have 6 additional covariates that are not used by the target function.

All numerical experiments were conducted using R-4.4.1 or Python-3.12 on a dedicated cluster with 2 Intel Xeon Gold 6302@2.1 GHz CPUs and 192 GB of memory. We modified the existing R package `glex` to compute the PD functions using `FastPD` and the path-dependent algorithm – which is due to (Friedman, 2001) but also reproduced as Algorithm 1 in (Lundberg et al., 2020). The code of the experiments can be found in the attached Git repository, `codeforsimulations`. Finally, we used `FastPD-100` to emulate the SHAP values that would have been computed by `TreeSHAP-int` since they are equivalent.

## A.1. Estimation Error Comparison - Figure 3

For this numerical experiment we generated iid datasets $\{(X^{(1)}, Y^{(1)}), \ldots, (X^{(n)}, Y^{(n)})\}$ over $B = 100$ repetitions with sample sizes $n = 500$ and $n = 5000$ from DGP 1. Multiple XGBoost models $(\hat{m}_n)$ were trained in each of the 100 repetitions with 5-fold cross-validation and their out-of-fold mean squared prediction error (MSPE) was computed. We ran the cross-validation with random search and 50 evaluations to tune the hyperparameters: `nrounds` $\in \{1, 2, \ldots, 1000\}$, `eta` $\in [0.01, 0.3]$ and `max_depth` $\in \{2, 3, \ldots, 6\}$. Following optimization, the best hyperparameter configuration was used to fit an XGBoost model on all $n$ observations, and the SHAP value $\phi_1$ was estimated for all $n$ observations using the different methods. The $n$ generated samples were also used as background samples. The SHAP MSEs were then computed as $\frac{1}{n} \sum_{i=1}^{n} (\phi_1^{\hat{m}_n}(X^{(i)}) - \hat{\phi}_1^{\hat{m}_n}(X^{(i)}))^2$, where $\hat{\phi}_1^{\hat{m}_n}$ is the estimate of the SHAP value $\phi_1^{\hat{m}_n}$ for the target function $\hat{m}_n$ for each method.

### A.1.1. ADDITIONAL SIMULATIONS ON FIGURE 3

We have furthermore conducted the same experiment with 8 covariates with pairwise correlation of 0 and 0.7, respectively. We notice that `TreeSHAP-path` performs worse in high-correlation scenarios, while our `FastPD` method is robust.

`TreeSHAP-path` can reliably estimate SHAP values when covariates are uncorrelated. However, its error remains higher than `FastPD` because it does not leverage all available samples. At each inner node, it conditions on the path taken by the evaluation point, significantly reducing the effective sample size. This effect is evident in the Figure 8 for $n = 500$.

## A.2. Inconsistency of `TreeSHAP-path` - Figure 2

For this numerical experiment, we followed the same procedure as in Section A.1. We selected the repetition where the MSE of `FastPD` matched its median MSE across all trials for the `FastPD` plot, and similarly, the repetition where `TreeSHAP-path` had median MSE for its plot.

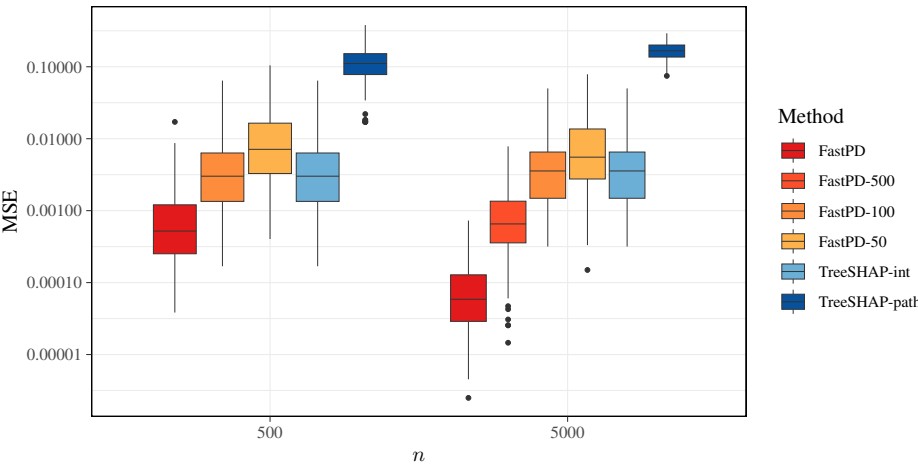

*Figure 7.* Reproduced Figure 3 with 8 covariates under the same target function as DGP 1, except that the covariates have a pairwise correlation of 0.7.

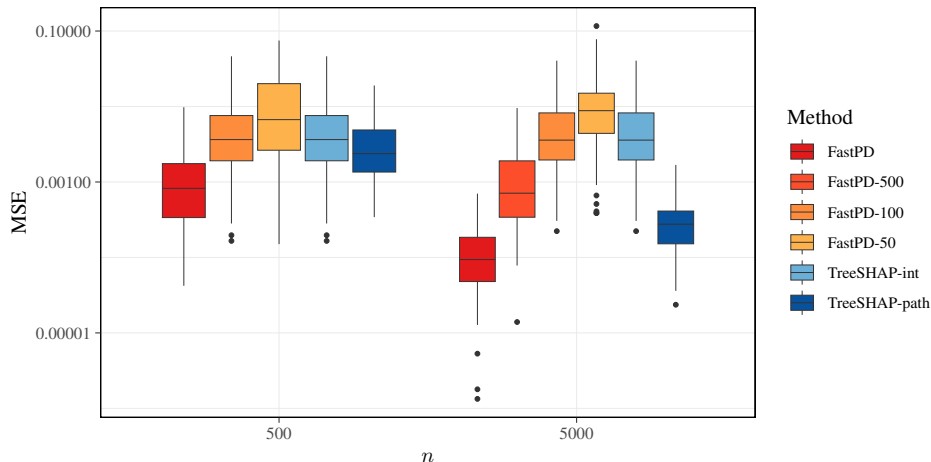

*Figure 8.* Reproduced Figure 3 with 8 covariates under the same target function as DGP 1, except that the covariates are uncorrelated.

### A.2.1. ADDITIONAL SIMULATIONS ON FIGURE 2

We also conducted the same experiment under DGP 2 with pairwise correlations of $\{0, 0.1, 0.3, 0.7\}$. The results indicate that `TreeSHAP-path` produces biased estimates of Model SHAP, whereas `FastPD` does not.

### A.3. Inconsistency of `Friedman-path` - Figure 5

For this numerical experiment we followed the same procedure as in Section A.1. We selected the single repetition for which the MSE of the `FastPD` $m_1$-component corresponded to the median MSE across all trials. In order to reproduce the `Friedman-path` plot, we used the R package `glex` which provided the path-dependent functional decomposition that was computed using the algorithm by Friedman (2001). Furthermore, we have modified the package to implement our FastPD algorithm, which we then used to compute the functional decomposition for the `FastPD` plot.

### A.3.1. ADDITIONAL SIMULATIONS ON FIGURE 5

We also conducted the same experiment under DGP 1 with pairwise correlations of 0.3 and 0.7. The median performing run is shown in Figure 10. We see that `FastPD` performs well in estimating the model functional component but by doing so it differs (and more so with increasing correlation) from the ground truth functional component.

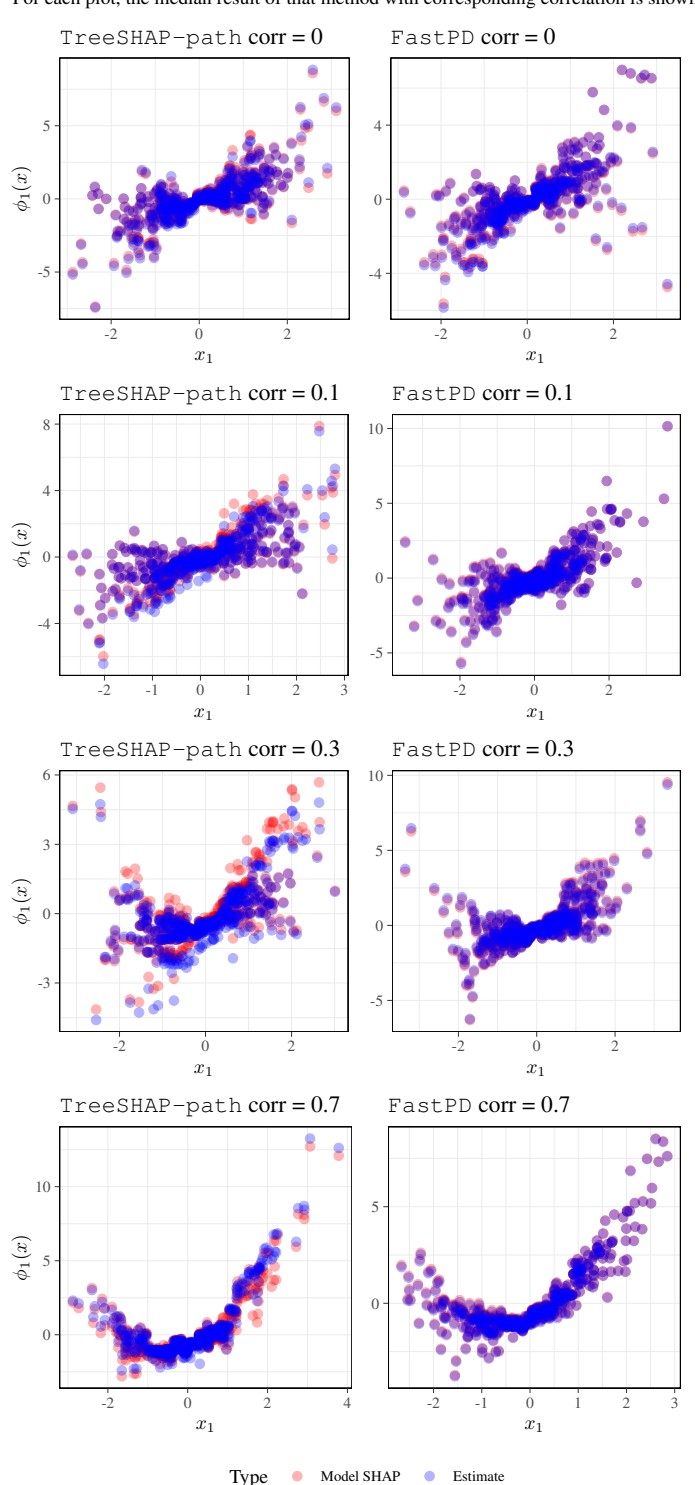

*Figure 9.* Reproduced Figure 2 with different pairwise correlations under DGP 2.

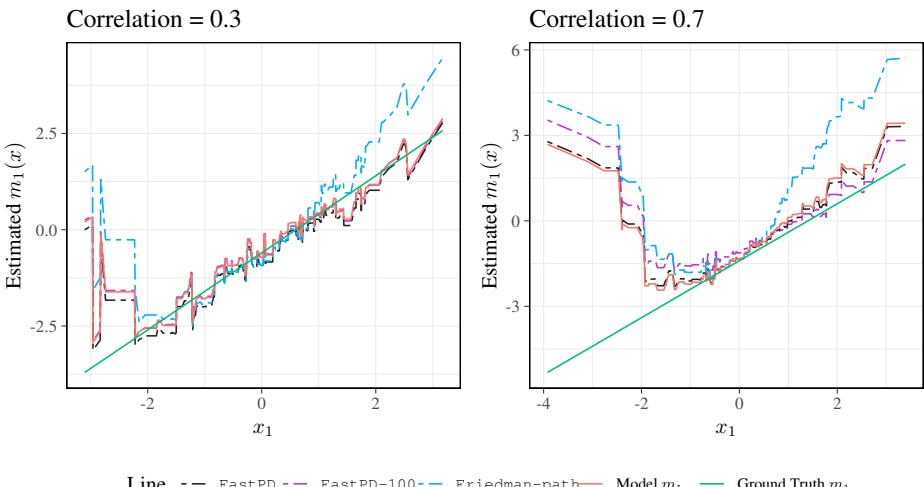

*Figure 10.* Plot of median `FastPD` and `Friedman-path` estimated component over 150 runs for correlations 0.3 and 0.7.

Figure 10 shows that in high-correlation settings the machine learning model can struggle to recover the true regression function; consequently, even though FastPD accurately estimates the model component, it does not target the underlying ground truth.

### A.4. Runtime Comparison - Figure 4

For this numerical experiment, we generated a single dataset of size $n = 8000$ using DGP 2 and fitted an XGBoost model with 20 trees and max-depth of 5, with the rest being the default XGBoost parameters. We performed no hyperparameter tuning here as we only wish to examine the runtime when $n$ is varied. Both the model $\hat{m}$ and the dataset was saved, we then evaluated the runtime as follows

1. For computing the functional components using `FastPD`: We implemented the algorithm in R (4.4.1) with Rcpp bindings to compute the all PD functions using `FastPD`.

2. For computing the SHAP values using `TreeSHAP-int`: We used Python-3.12 and modified the `shap` package[¶] to compute the SHAP explanations for all features using arbitrary many background samples.

3. For computing the SHAP values using Zern et al. (2023): We used Python 3.12 and the `pltreeshap` package [‖] to compute the SHAP explanations for all features using arbitrary many background samples

4. For computing the SHAP values using `VanillaPD`: We wrote a simple script in Python 3.12 which repeatedly calls the predict function of XGBoost on the synthetic samples created from the background samples and the evaluation points, the predictions are then averaged to obtain the partial dependence functions.

For all $k \in \{1000, 2000, \ldots, 8000\}$, we took a subset $\mathcal{D}$ of the original dataset of size $k$ and used it both as background and evaluation data (i.e., $n_b = n_f = k$). We then ran all methods 100 times to obtain SHAP values for $n_e$ evaluation points using $n_b$ background samples.

## B. Evaluation Algorithm

---

[¶]GitHub: `https://github.com/shap/shap/`

[‖]GitHub: `https://github.com/schufa-innovationlab/pltreeshap/tree/main`

---

**Algorithm 2** `FastPD` evaluation step to calculate $\hat{v}_S(x_S)$. To be applied after augmenting the tree as in Algorithm 1.

---

**input** Query point $x$, feature set $S$, tree structure, path features $T$, path data $P$
**output** $\hat{v}_S(x_S)$ – PD-function evaluated on $x_S$

1: $U \leftarrow S \cap \left( \bigcup_j T_j \right)$
2: **if** $\hat{v}_U(x_U)$ calculated before **then**
3:     **return** $\hat{v}_U(x_U)$
4: **end if**
5: **function** G$(j)$
6:     **if** $j$ is leaf **then**
7:         $U_j \leftarrow U \cap T_j$
8:         Extract $D_{U_j}^{(j)}$ from $P_j$
9:         $\hat{P} \leftarrow \text{length}(D_{U_j}^{(}j))/n_b$
10:         **return** $v_j \cdot \hat{P}$
11:     **end if**
12:     **if** $d_j \in U$ **then**
13:         **if** $x_{d_j} \leq t_j$ **then**
14:             **return** G$(l_j)$
15:         **else**
16:             **return** G$(r_j)$
17:         **end if**
18:     **else**
19:         **return** G$(l_j)$ + G$(r_j)$
20:     **end if**
21: **end function**
22: $\hat{v}_U(x_U) \leftarrow$ G$(1)$
23: $\hat{v}_S(x_S) \leftarrow \hat{v}_U(x_U)$

---

## C. Proofs

### C.1. Proof of Proposition 3.1

*Proof.* We show (i) and (ii) via an example, let $P_X$ be a distribution over $X = (X_1, X_2) \in \mathbb{R}^2$ such that

$$P_X(X = x) = \begin{cases} 500/2500 = 0.2 & \text{if } x = (0,0), \\ 250/2500 = 0.1 & \text{if } x = (0, 0.4), \\ 250/2500 = 0.1 & \text{if } x = (0.7, 0), \\ 1500/2500 = 0.6 & \text{if } x = (0.7, 0.4), \\ 0 & \text{otherwise.} \end{cases}$$

Next, assume $n_b = 2500$ observations sampled from from $P_X$. There is a non-zero probability that $\mathcal{D}_{n_b}$ satisfies

$$\sum_{i=1}^{2500} \mathbb{1}(X^{(i)} = x) = \begin{cases} 500 & \text{if } x = (0,0), \\ 250 & \text{if } x = (0, 0.4), \\ 250 & \text{if } x = (0.7, 0), \\ 1500 & \text{if } x = (0.7, 0.4) \\ 0 & \text{otherwise.} \end{cases}$$

We now consider two decision trees $\hat{m}^A$ and $\hat{m}^B$ as depicted in Figure 11. The leaves, going from left to right, are labeled $L_1$ to $L_4$ and are identical for both trees, implying that they are functionally equivalent, i.e., $\hat{m}^A(x) = \hat{m}^B(x)$ for all $x$.

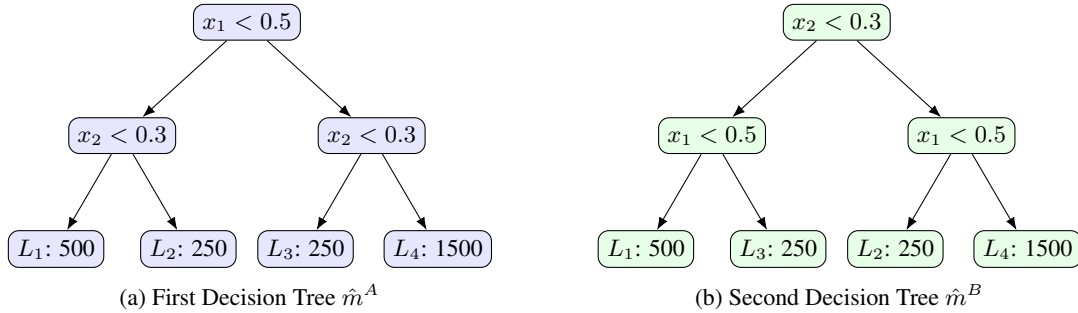

(a) First Decision Tree $\hat{m}^A$        (b) Second Decision Tree $\hat{m}^B$

*Figure 11.* The two trees have the same leaves hence predict the same values, but their explanations differ when obtained via `TreeSHAP-path`. The number on each leaf is the number of observations landing in that leaf. The left-branch is followed when the split condition is true.

We first prove $(i)$ by showing that their SHAP values differ when they are computed using the path-dependent algorithm on $\mathcal{D}_{n_b}$. Indeed, let $x = (0.1, 0.2)$ be the observation to be explained and $V_N$ be the value of leaf $N$. We follow the left branch if the split condition is satisfied. Assume that $V_1 = 10, V_2 = -5, V_3 = -5$ and $V_4 = 10$. In the following denote by $\tilde{v}^{\hat{m}^A}$ and $\tilde{v}^{\hat{m}^B}$ the estimates of the path-dependent PD functions (used in both `TreeSHAP-path` and `Friedman-path`). For the first tree, $\tilde{v}_S^{\hat{m}^A}(x_S)$, using $\mathcal{D}_{n_b}$ equals

$$\tilde{v}_\emptyset^{\hat{m}^A} = \frac{1}{2500}(500 \cdot V_1 + 250 \cdot V_2 + 250 \cdot V_3 + 1500 \cdot V_4) = 7,$$

$$\tilde{v}_1^{\hat{m}^A}(x_1) = \frac{1}{750}(500 \cdot V_1 + 250 \cdot V_2) = 5,$$

$$\tilde{v}_2^{\hat{m}^A}(x_2) = \frac{1}{2500}(750 \cdot V_1 + 1750 \cdot V_3) = -0.5,$$

$$\tilde{v}_{1,2}^{\hat{m}^A}(x_1, x_2) = V_1 = 10.$$

For the second tree, $\tilde{v}_S^{\hat{m}^B}(x_S)$, using $\mathcal{D}_{n_b}$ equals

$$\tilde{v}_\emptyset^{\hat{m}^B} = \frac{1}{2500}(500 \cdot V_1 + 250 \cdot V_3 + 250 \cdot V_2 + 1500 \cdot V_4) = 7,$$

$$\tilde{v}_1^{\hat{m}^B}(x_1) = \frac{1}{2500}(750 \cdot V_1 + 1750 \cdot V_2) = -0.5,$$

$$\tilde{v}_2^{\hat{m}^B}(x_2) = \frac{1}{750}(500 \cdot V_1 + 250 \cdot V_3) = 5,$$

$$\tilde{v}_{1,2}^{\hat{m}^B}(x_1, x_2) = V_1 = 10.$$

Finally, the `TreeSHAP-path` estimates of the SHAP value for feature $x_1$ in both trees are given as

$$\hat{\phi}_1^{\hat{m}^A} = \frac{1}{2}(\tilde{v}_{1,2}^{\hat{m}^A}(x_1, x_2) - \tilde{v}_2^{\hat{m}^A}(x_2) + \tilde{v}_1^{\hat{m}^A}(x_1) - \tilde{v}_\emptyset^{\hat{m}^A}) = 4.25,$$

$$\hat{\phi}_1^{\hat{m}^B} = \frac{1}{2}(\tilde{v}_{1,2}^{\hat{m}^B}(x_1, x_2) - \tilde{v}_2^{\hat{m}^B}(x_2) + \tilde{v}_1^{\hat{m}^B}(x_1) - \tilde{v}_\emptyset^{\hat{m}^B}) = -1.25.$$

The computed SHAP values not only differ but have opposite signs! We observe that for $\hat{m}^A$, feature $x_1$ has a positive attribution, whereas the attribution is negative for $\hat{m}^B$. We can also compute the empirical PD functions for both trees as follows

$$\hat{v}_\emptyset = \frac{1}{2500}(500 \cdot V_1 + 250 \cdot V_2 + 250 \cdot V_3 + 1500 \cdot V_4) = 7,$$

$$\hat{v}_1(x_1) = \frac{1}{2500}(750 \cdot V_1 + 1750 \cdot V_2) = -0.5,$$

$$\hat{v}_2(x_2) = \frac{1}{2500}(750 \cdot V_1 + 1750 \cdot V_3) = -0.5,$$

$$\hat{v}_{1,2}(x_1, x_2) = V_1 = 10.$$

And thus, the empirical SHAP estimate is given as

$$\hat{\phi}_1 = \frac{1}{2}(\hat{v}_{1,2}^{\hat{m}^A}(x_1, x_2) - \hat{v}_2^{\hat{m}^A}(x_2) + \hat{v}_1^{\hat{m}^A}(x_1) - \hat{v}_\emptyset^{\hat{m}^A}) = 1.5,$$

which is the same for both $\hat{m}^A$ and $\hat{m}^B$. Furthermore, by construction, the empirical SHAP estimate is equal to the population SHAP value, $\hat{\phi}_1 = \phi_1^{\hat{m}^A} = \phi_1^{\hat{m}^B}$.

We now show $(ii)$. Let $\#L_N$ denote the number of observations that fall in leaf $N$. The path-dependent approximations of the PD functions for the first tree can be alternatively written as

$$\tilde{v}_\emptyset^{\hat{m}^A} = \frac{1}{n_b}(\#L_1 \cdot V_1 + \#L_2 \cdot V_2 + \#L_3 \cdot V_3 + \#L_4 \cdot V_4),$$

$$\tilde{v}_1^{\hat{m}^A}(x_1) = \frac{1}{\#L_1 + \#L_2}(\#L_1 \cdot V_1 + \#L_2 \cdot V_2),$$

$$\tilde{v}_2^{\hat{m}^A}(x_2) = \frac{1}{n_b}((\#L_1 + \#L_2) \cdot V_1 + (\#L_3 + \#L_4) \cdot V_3),$$

$$\tilde{v}_{1,2}^{\hat{m}^A}(x_1, x_2) = V_1 = 10.$$

By the strong law of large numbers it holds that $\#L_N/n_b \xrightarrow{a.s.} P_X(X \in L_N)$ as $n_b \longrightarrow \infty$, so therefore $\tilde{v}_\emptyset^{\hat{m}^A} \xrightarrow{a.s.} 7$ and $\tilde{v}_2^{\hat{m}^A}(x_2) \xrightarrow{a.s.} -0.5$. However, since $(\#L_1 + \#L_2)/n_b \xrightarrow{a.s.} P_X(X \in L_1) + P_X(X \in L_2)$ we have the following

$$\frac{\#L_1}{\#L_1 + \#L_2} \xrightarrow{a.s.} \frac{P_X(X \in L_1)}{P_X(X \in L_1) + P_X(X \in L_2)} = 0.2/0.3 = 500/750,$$

$$\frac{\#L_2}{\#L_1 + \#L_2} \xrightarrow{a.s.} \frac{P_X(X \in L_2)}{P_X(X \in L_1) + P_X(X \in L_2)} = 0.1/0.3 = 250/750.$$

Hence $\tilde{v}_1^{\hat{m}^A}(x_1) \xrightarrow{a.s.} 5$, implying that $\hat{\phi}_1^{\hat{m}^A} \xrightarrow{a.s.} 4.25$, which is not the same as the population SHAP, which was 1.5. $\quad\square$

## C.2. Proof of Proposition 3.2

*Proof.* First, observe that the PD function $v_S^m(x_S) = \mathbb{E}_{P_X}[m(x_S, X_{\bar{S}})]$ of $m$ exists with respect to any $S \subseteq [d]$ since $m$ is bounded.

For the first part of the statement, we fix $S \subseteq [d]$. Since `FastPD` exactly evaluates the empirical PD function, it holds for all $x \in \mathcal{X}$ that

$$\hat{v}_{S,n_b}^m(x_S) = \frac{1}{n_b} \sum_{i=1}^{n_b} m(x_S, X_{\bar{S}}^{(i)}).$$

Therefore, since $m$ is bounded the strong law of large numbers implies that

$$\hat{v}_{S,n_b}^m \xrightarrow{a.s.} v_S^m(x_S) \qquad \text{for } n_b \longrightarrow \infty.$$

For the second part of the statement, again fix $S \subseteq [d]$ and let $\hat{m}_{n_b}$ be a uniformly consistent estimate of $m$ trained on $\mathcal{D}_{n_b}$ observations. Then by applying the triangle inequality it readily follows for all $x \in \mathcal{X}$ that

$$\left| v_{S,n_b}^{\hat{m}_{n_b}}(x_S) - v_S^m(x_S) \right| = \left| \frac{1}{n_b} \sum_{i=1}^{n_b} \hat{m}_{n_b}(x_S, X_{\bar{S}}^{(i)}) - \mathbb{E}_{P_X}[m(x_S, X_{\bar{S}})] \right|$$

$$= \left| \frac{1}{n_b} \sum_{i=1}^{n_b} \hat{m}_{n_b}(x_S, X_{\bar{S}}^{(i)}) - \frac{1}{n_b} \sum_{i=1}^{n_b} m(x_S, X_{\bar{S}}^{(i)}) + \frac{1}{n_b} \sum_{i=1}^{n_b} m(x_S, X_{\bar{S}}^{(i)}) - \mathbb{E}_{P_X}[m(x_S, X_{\bar{S}})] \right|$$

$$\leq \frac{1}{n_b} \sum_{i=1}^{n_b} \left| \hat{m}_{n_b}(x_S, X_{\bar{S}}^{(i)}) - m(x_S, X_{\bar{S}}^{(i)}) \right| + \left| \frac{1}{n_b} \sum_{i=1}^{n_b} m(x_S, X_{\bar{S}}^{(i)}) - \mathbb{E}_{P_X}[m(x_S, X_{\bar{S}})] \right|$$

$$\leq \frac{1}{n_b} \sum_{i=1}^{n_b} \sup_x |\hat{m}_{n_b}(x) - m(x)| + \left| \frac{1}{n_b} \sum_{i=1}^{n_b} m(x_S, X_{\bar{S}}^{(i)}) - \mathbb{E}_{P_X}[m(x_S, X_{\bar{S}})] \right| \xrightarrow{a.s.} 0,$$

where the convergence follows from using uniform consistency of $\hat{m}_{n_b}$ and the consistency of the empirical PD function which follows from the strong law of large numbers as above. $\quad\square$

## D. Experiments on Real Data

### D.1. Adult Dataset - Figure 6

To reproduce Figure 6, we used the `Adult` dataset where we first removed all observations with missing values and applied ordinal encoding to every categorical feature. We then performed a randomized grid search (25 evaluations) with 5-fold cross-validation to tune the following XGBoost hyperparameters:

$$\texttt{max\_depth} \in \{3, 4, 5, 6, 7\}, \quad \texttt{eta} \in \{0.01, 0.05, 0.1\},$$
$$\texttt{nrounds} \in \{100, 200, 300\}, \quad \texttt{min\_child\_weight} \in \{1, 3, 5\},$$
$$\texttt{subsample} \in \{0.6, 0.8, 1.0\}, \quad \texttt{colsample\_bytree} \in \{0.6, 0.8, 1.0\}.$$

The best-performing model had the following hyperparameters:

$$\texttt{subsample} = 0.8, \quad \texttt{min\_child\_weight} = 1,$$
$$\texttt{max\_depth} = 6, \quad \texttt{eta} = 0.05,$$
$$\texttt{colsample\_bytree} = 0.6, \quad \texttt{nrounds} = 100$$

### D.2. FastPD vs Path-dependent (Regression)

We summarize the results of the experiments comparing `FastPD` with path-dependent methods in the regression setting below. We used the OpenML-CTR23 Regression Task Collection, which contains 35 varied regression datasets. On each dataset, we ran 5-fold cross-validation, tuning the following hyperparameters for the XGBoost model over 100 randomly sampled settings:

$$\texttt{max\_depth} \in [1, 5], \quad \texttt{eta} \in [0, 0.5],$$
$$\texttt{nrounds} \in [10, 200], \quad \texttt{colsample\_bytree} \in [0.5, 1],$$
$$\texttt{subsample} \in [0.5, 1].$$

The best-performing model was then selected based on the cross-validated MSE, and used by `FastPD`, `FastPD (50)`, `FastPD (100)` and the path-dependent method (`Friedman--path`) to compute the functional components (see Figure 5). We then applied the variable-importance measure for each component $S$ as

$$\text{Importance}(\widehat{m}_S) = \frac{1}{n} \sum_{i=1}^{n} |\widehat{m}_S(X_S^{(i)})|.$$

This enables us to rank the importance of each component. We present the importances of all components that were ranked in the top five by *any* method for each dataset. Using `FastPD` as the reference method, the relative differences

$$\frac{\text{Importance(other)} - \text{Importance(\texttt{FastPD})}}{\text{Importance(\texttt{FastPD})}}$$

of each method with respect to `FastPD` are reported in parentheses. Finally, for each dataset we report in parentheses the number of observations $n$, the feature dimension $p$ and the maximum depth of the XGBoost model $D$. Note that `FastPD` and the path-dependent method are trivially equivalent for those datasets where the tuned XGBoost model had $D = 1$.

| Dataset | Variable/Interaction | FastPD (Reference) | FastPD (50) | FastPD (100) | Path-dependent |
|---|---|---|---|---|---|
| Moneyball (n: 1232, p: 15, D: 2) | Intercept | **713.8** | 694.4 (−2.7%) | 715.5 (+0.2%) | 713.8 (0%) |
| | SLG | **35.96** | 35.91 (−0.1%) | 36.12 (+0.4%) | 37.11 (+3.2%) |
| | OBP | **28.36** | 28.45 (+0.3%) | 28.2 (−0.6%) | 28.5 (+0.5%) |
| | W | **15.26** | 15.94 (+4.5%) | 15.52 (+1.7%) | 13.03 (−14.6%) |
| | RA | **10.95** | 11.05 (+0.9%) | 11.4 (+4.1%) | 10.51 (−4%) |
| QSAR_fish_toxicity (n: 908, p: 7, D: 5) | Intercept | **4.066** | 4.143 (+1.9%) | 4.125 (+1.5%) | 4.066 (0%) |
| | MLOGP | **0.4468** | 0.4085 (−8.6%) | 0.4514 (+1%) | 0.5795 (+29.7%) |
| | SM1_Dz | **0.4279** | 0.4339 (+1.4%) | 0.4323 (+1%) | 0.4099 (−4.2%) |
| | GATS1i | **0.2743** | 0.2865 (+4.4%) | 0.2869 (+4.6%) | 0.2568 (−6.4%) |
| | CIC0 | **0.1907** | 0.1681 (−11.9%) | 0.2047 (+7.3%) | 0.1703 (−10.7%) |
| abalone (n: 4177, p: 9, D: 5) | Intercept | **9.934** | 10.08 (+1.5%) | 10.17 (+2.4%) | 9.934 (0%) |
| | shell_weight | **1.388** | 1.33 (−4.2%) | 1.358 (−2.2%) | 1.206 (−13.1%) |
| | shucked_weight | **1.26** | 1.349 (+7.1%) | 1.311 (+4%) | 0.8631 (−31.5%) |
| | shucked_weight:whole_weight | **0.8577** | 0.8526 (−0.6%) | 0.8585 (+0.1%) | 0.4454 (−48.1%) |
| | whole_weight | **0.7164** | 0.7108 (−0.8%) | 0.6996 (−2.3%) | 0.4368 (−39%) |
| airfoil_self_noise (n: 1503, p: 6, D: 5) | Intercept | **124.8** | 124.9 (+0.1%) | 125.1 (+0.2%) | 124.8 (0%) |
| | frequency | **3.335** | 3.389 (+1.6%) | 3.554 (+6.6%) | 2.734 (−18%) |
| | displacement_thickness:frequency | **2.985** | 2.841 (−4.8%) | 2.719 (−8.9%) | 2.521 (−15.5%) |
| | chord_length:frequency | **1.837** | 1.963 (+6.9%) | 1.93 (+5.1%) | 1.821 (−0.9%) |
| | chord_length | **1.678** | 1.637 (−2.4%) | 1.998 (+19.1%) | 1.543 (−8%) |
| auction_verification (n: 2043, p: 8, D: 5) | Intercept | **7337** | 7932 (+8.1%) | 8331 (+13.5%) | 7337 (0%) |
| | process.b1.capacity | **4712** | 5548 (+17.7%) | 4625 (−1.8%) | 4571 (−3%) |
| | property.product.2 | **3318** | 3479 (+4.9%) | 3374 (+1.7%) | 4126 (+24.4%) |
| | process.b1.capacity:property.product.2 | **2329** | 2642 (+13.4%) | 2176 (−6.6%) | 2627 (+12.8%) |
| | property.product.6 | **2219** | 2029 (−8.6%) | 1960 (−11.7%) | 1667 (−24.9%) |
| brazilian_houses (n: 10692, p: 10, D: 1) | Intercept | **5487** | 5821 (+6.1%) | 4867 (−11.3%) | 5487 (0%) |
| | hoa | **1718** | 1791 (+4.2%) | 1608 (−6.4%) | 1718 (0%) |
| | area | **1320** | 1364 (+3.3%) | 1282 (−2.9%) | 1320 (0%) |
| | bathroom | **620.8** | 576.2 (−7.2%) | 595 (−4.2%) | 620.8 (0%) |
| | furniture.not.furnished | **312.1** | 336.5 (+7.8%) | 314.8 (+0.9%) | 312.1 (0%) |
| california_housing (n: 20640, p: 9, D: 5) | Intercept | **206800** | 207700 (+0.4%) | 218700 (+5.8%) | 206800 (0%) |
| | latitude | **93530** | 90120 (−3.6%) | 90150 (−3.6%) | 53530 (−42.8%) |
| | latitude:longitude | **82130** | 85870 (+4.6%) | 83530 (+1.7%) | 48730 (−40.7%) |

| | | | | | |
|---|---|---|---|---|---|
| | longitude | **75310** | 77420 (+2.8%) | 80700 (+7.2%) | 52040 (−30.9%) |
| | medianIncome | **42210** | 40590 (−3.8%) | 43190 (+2.3%) | 48430 (+14.7%) |
| cars (n: 804, p: 18, D: 5) | Intercept | **21330** | 19870 (−6.8%) | 23490 (+10.1%) | 21330 (0%) |
| | Cylinder | **2701** | 2688 (−0.5%) | 3154 (+16.8%) | 2869 (+6.2%) |
| | Cadillac | **2696** | 1775 (−34.2%) | 3285 (+21.8%) | 2604 (−3.4%) |
| | Saab | **2271** | 2251 (−0.9%) | 2231 (−1.8%) | 2406 (+5.9%) |
| | Mileage | **1166** | 1152 (−1.2%) | 1213 (+4%) | 1164 (−0.2%) |
| concrete_compressive_strength (n: 1030, p: 9, D: 4) | Intercept | **35.82** | 38.19 (+6.6%) | 36.2 (+1.1%) | 35.82 (0%) |
| | age | **8.087** | 8.365 (+3.4%) | 7.96 (−1.6%) | 7.641 (−5.5%) |
| | cement | **6.263** | 6.089 (−2.8%) | 6.238 (−0.4%) | 5.942 (−5.1%) |
| | water | **4.093** | 4.145 (+1.3%) | 4.104 (+0.3%) | 3.914 (−4.4%) |
| | blast_furnace_slag | **2.702** | 2.623 (−2.9%) | 2.722 (+0.7%) | 2.429 (−10.1%) |
| cps88wages (n: 28155, p: 7, D: 4) | Intercept | **604** | 546.8 (−9.5%) | 628 (+4%) | 604 (0%) |
| | education | **122.3** | 109 (−10.9%) | 128.4 (+5%) | 125.5 (+2.6%) |
| | experience | **119.3** | 121.1 (+1.5%) | 123.8 (+3.8%) | 114.5 (−4%) |
| | parttime.yes | **45.72** | 65.15 (+42.5%) | 43.05 (−5.8%) | 46.77 (+2.3%) |
| | smsa.yes | **36.99** | 33.19 (−10.3%) | 39.36 (+6.4%) | 37.16 (+0.5%) |
| cpu_activity (n: 8192, p: 22, D: 5) | Intercept | **83.97** | 83.81 (−0.2%) | 84.37 (+0.5%) | 83.97 (0%) |
| | freeswap | **5.133** | 4.076 (−20.6%) | 4.725 (−7.9%) | 5.126 (−0.1%) |
| | vflt | **2.359** | 2.58 (+9.4%) | 2.325 (−1.4%) | 3.304 (+40.1%) |
| | scall | **1.597** | 1.808 (+13.2%) | 1.708 (+7%) | 1.687 (+5.6%) |
| | pflt | **0.8853** | 0.9213 (+4.1%) | 0.959 (+8.3%) | 0.8976 (+1.4%) |
| diamonds (n: 53940, p: 10, D: 5) | Intercept | **3933** | 3949 (+0.4%) | 4278 (+8.8%) | 3933 (0%) |
| | carat | **2329** | 2322 (−0.3%) | 2400 (+3%) | 2250 (−3.4%) |
| | y | **1242** | 1351 (+8.8%) | 1289 (+3.8%) | 1616 (+30.1%) |
| | clarity.VS2 | **407.1** | 418.2 (+2.7%) | 374.4 (−8%) | 300.5 (−26.2%) |
| | clarity.VS1 | **363.3** | 303.3 (−16.5%) | 294 (−19.1%) | 274.3 (−24.5%) |
| | x:y | **308.6** | 363.6 (+17.8%) | 421.5 (+36.6%) | 129.2 (−58.1%) |
| | carat:y | **293.8** | 318.2 (+8.3%) | 308.1 (+4.9%) | 644.9 (+119.5%) |
| | clarity.VVS1 | **243.8** | 410.6 (+68.4%) | 248.8 (+2.1%) | 181 (−25.8%) |
| energy_efficiency (n: 768, p: 9, D: 5) | Intercept | **22.31** | 21.34 (−4.3%) | 22.67 (+1.6%) | 22.31 (0%) |
| | overall_height | **10.07** | 10.05 (−0.2%) | 10.56 (+4.9%) | 8.534 (−15.3%) |
| | glazing_area | **2.17** | 2.326 (+7.2%) | 2.044 (−5.8%) | 2.164 (−0.3%) |
| | overall_height:relative_compactness | **1.592** | 1.585 (−0.4%) | 1.386 (−12.9%) | 1.6 (+0.5%) |

| Dataset | Feature | Value | | | |
|---|---|---|---|---|---|
| | relative_compactness | **1.495** | 1.494 (−0.1%) | 1.152 (−22.9%) | 1.561 (+4.4%) |
| fifa (n: 19178, p: 29, D: 3) | Intercept | **9020** | 15910 (+76.4%) | 9309 (+3.2%) | 9020 (0%) |
| | overall | **7661** | 11650 (+52.1%) | 7544 (−1.5%) | 7775 (+1.5%) |
| | skill_ball_control | **884.3** | 955.9 (+8.1%) | 1112 (+25.7%) | 899.5 (+1.7%) |
| | nationality_name.England | **522.7** | 968.5 (+85.3%) | 497.6 (−4.8%) | 494.1 (−5.5%) |
| | attacking_heading_accuracy | **426.7** | 448.7 (+5.2%) | 377.7 (−11.5%) | 399.3 (−6.4%) |
| | skill_dribbling | **250.7** | 441.7 (+76.2%) | 412.6 (+64.6%) | 229.7 (−8.4%) |
| | defending_standing_tackle | **237.1** | 710.9 (+199.8%) | 334.9 (+41.2%) | 262.1 (+10.5%) |
| forest_fires (n: 517, p: 13, D: 2) | Intercept | **3.536** | 5.088 (+43.9%) | 4.289 (+21.3%) | 3.536 (0%) |
| | temp | **1.184** | 2.376 (+100.7%) | 1.748 (+47.6%) | 1.218 (+2.9%) |
| | DMC | **0.3103** | 0.3302 (+6.4%) | 0.3246 (+4.6%) | 0.2826 (−8.9%) |
| | day.sat:temp | **0.1736** | 0.1705 (−1.8%) | 0.1967 (+13.3%) | 0.1701 (−2%) |
| | Y | **0.1575** | 0.1124 (−28.6%) | 0.1384 (−12.1%) | 0.1575 (0%) |
| | day.sat | **0.1539** | 0.1403 (−8.8%) | 0.1786 (+16%) | 0.1499 (−2.6%) |
| fps_benchmark (n: 24624, p: 44, D: 4) | Intercept | **123.6** | 113 (−8.6%) | 121.5 (−1.7%) | 123.6 (0%) |
| | GameSetting.max | **15.59** | 15.16 (−2.8%) | 15.67 (+0.5%) | 15.59 (0%) |
| | GameName.battlefield4 | **9.474** | 11.48 (+21.2%) | 8.28 (−12.6%) | 9.267 (−2.2%) |
| | GameName.worldOfTanks | **9.219** | 6.357 (−31%) | 5.871 (−36.3%) | 8.987 (−2.5%) |
| | GameName.totalWar3Kingdoms | **8.719** | 10.05 (+15.3%) | 7.265 (−16.7%) | 8.491 (−2.6%) |
| | GameName.grandTheftAuto5 | **8.142** | 6.011 (−26.2%) | 11.18 (+37.3%) | 7.957 (−2.3%) |
| | GpuNumberOfTransistors | **7.022** | 6.575 (−6.4%) | 7.271 (+3.5%) | 8.611 (+22.6%) |
| | GameName.apexLegends | **6.837** | 16.53 (+141.8%) | 9.649 (+41.1%) | 6.594 (−3.6%) |
| | GameName.destiny2 | **5.488** | 11.93 (+117.4%) | 6.69 (+21.9%) | 5.311 (−3.2%) |
| geographical_origin_of_music (n: 1059, p: 117, D: 4) | Intercept | **26.68** | 26.11 (−2.1%) | 29.12 (+9.1%) | 26.68 (0%) |
| | V32 | **2.66** | 2.762 (+3.8%) | 2.486 (−6.5%) | 2.921 (+9.8%) |
| | V92 | **2.016** | 2.229 (+10.6%) | 2.013 (−0.1%) | 1.627 (−19.3%) |
| | V4 | **1.53** | 1.787 (+16.8%) | 1.46 (−4.6%) | 1.288 (−15.8%) |
| | V90 | **1.392** | 1.287 (−7.5%) | 1.311 (−5.8%) | 1.469 (+5.5%) |
| | V91 | **1.173** | 1.358 (+15.8%) | 0.9099 (−22.4%) | 1.141 (−2.7%) |
| grid_stability (n: 10000, p: 13, D: 5) | Intercept | **0.01574** | 0.01094 (−30.5%) | 0.01687 (+7.2%) | 0.01574 (0%) |
| | tau2 | **0.01023** | 0.01183 (+15.6%) | 0.009302 (−9.1%) | 0.01034 (+1.1%) |
| | tau4 | **0.01021** | 0.008786 (−13.9%) | 0.008845 (−13.4%) | 0.01017 (−0.4%) |
| | tau3 | **0.01014** | 0.008657 (−14.6%) | 0.0108 (+6.5%) | 0.01014 (0%) |
| | tau1 | **0.01008** | 0.01042 (+3.4%) | 0.01186 (+17.7%) | 0.01015 (+0.7%) |

| | | | | | |
|---|---|---|---|---|---|
| | g3 | **0.009318** | 0.009932 (+6.6%) | 0.009094 (−2.4%) | 0.009422 (+1.1%) |
| | g2 | **0.009225** | 0.00834 (−9.6%) | 0.01086 (+17.7%) | 0.009254 (+0.3%) |
| | g4 | **0.009084** | 0.009958 (+9.6%) | 0.009448 (+4%) | 0.009025 (−0.6%) |
| health_insurance (n: 22272, p: 12, D: 5) | Intercept | **25.5** | 24.86 (−2.5%) | 27.17 (+6.5%) | 25.5 (0%) |
| | whi.yes | **8.041** | 8.252 (+2.6%) | 7.856 (−2.3%) | 8.687 (+8%) |
| | experience | **3.469** | 3.612 (+4.1%) | 3.239 (−6.6%) | 3.092 (−10.9%) |
| | kidslt6 | **2.557** | 2.594 (+1.4%) | 2.497 (−2.3%) | 2.142 (−16.2%) |
| | husby | **1.637** | 1.711 (+4.5%) | 1.434 (−12.4%) | 1.636 (−0.1%) |
| kin8nm (n: 8192, p: 9, D: 5) | Intercept | **0.714** | 0.7176 (+0.5%) | 0.7396 (+3.6%) | 0.714 (0%) |
| | theta3 | **0.1264** | 0.1068 (−15.5%) | 0.1331 (+5.3%) | 0.1253 (−0.9%) |
| | theta5 | **0.05947** | 0.06424 (+8%) | 0.05806 (−2.4%) | 0.05835 (−1.9%) |
| | theta4:theta7 | **0.04003** | 0.04405 (+10%) | 0.03762 (−6%) | 0.0403 (+0.7%) |
| | theta4:theta6 | **0.03859** | 0.04127 (+6.9%) | 0.03474 (−10%) | 0.03879 (+0.5%) |
| | theta6 | **0.03624** | 0.02494 (−31.2%) | 0.04615 (+27.3%) | 0.03586 (−1%) |
| | theta7 | **0.03544** | 0.03077 (−13.2%) | 0.04927 (+39%) | 0.03534 (−0.3%) |
| kings_county (n: 21613, p: 22, D: 5) | Intercept | **540100** | 548000 (+1.5%) | 586000 (+8.5%) | 540100 (0%) |
| | lat | **106400** | 104800 (−1.5%) | 109600 (+3%) | 105700 (−0.7%) |
| | sqft_living | **64580** | 65630 (+1.6%) | 67860 (+5.1%) | 74220 (+14.9%) |
| | grade | **61040** | 59060 (−3.2%) | 70030 (+14.7%) | 75760 (+24.1%) |
| | long | **28700** | 27900 (−2.8%) | 35610 (+24.1%) | 27830 (−3%) |
| miami_housing (n: 13932, p: 16, D: 4) | Intercept | **399900** | 436400 (+9.1%) | 410000 (+2.5%) | 399900 (0%) |
| | TOT_LVG_AREA | **66800** | 69390 (+3.9%) | 67920 (+1.7%) | 82580 (+23.6%) |
| | OCEAN_DIST | **39470** | 42860 (+8.6%) | 36060 (−8.6%) | 48600 (+23.1%) |
| | CNTR_DIST | **37680** | 49020 (+30.1%) | 42160 (+11.9%) | 39730 (+5.4%) |
| | SUBCNTR_DI | **35880** | 37150 (+3.5%) | 36230 (+1%) | 34710 (−3.3%) |
| | LONGITUDE | **34240** | 37200 (+8.6%) | 32940 (−3.8%) | 25080 (−26.8%) |
| naval_propulsion_plant (n: 11934, p: 15, D: 5) | Intercept | **0.975** | 0.9752 (0%) | 0.9745 (−0.1%) | 0.975 (0%) |
| | hp_turbine_exit_pressure | **0.01705** | 0.01745 (+2.3%) | 0.01698 (−0.4%) | 0.007992 (−53.1%) |
| | gt_compressor_outlet_air_temperature | **0.01571** | 0.01587 (+1%) | 0.0157 (−0.1%) | 0.01388 (−11.6%) |
| | gas_turbine_shaft_torque | **0.005027** | 0.004993 (−0.7%) | 0.005038 (+0.2%) | 0.003613 (−28.1%) |
| | gas_turbine_exhaust_gas_pressure | **0.004985** | 0.005062 (+1.5%) | 0.005043 (+1.2%) | 0.003119 (−37.4%) |
| | gt_compressor_outlet_air_temperature:hp_turbine_exit_pressure | **0.003217** | 0.003212 (−0.2%) | 0.003262 (+1.4%) | 0.004906 (+52.5%) |
| physiochemical_protein (n: 45730, p: 10, D: 5) | Intercept | **7.749** | 6.979 (−9.9%) | 7.72 (−0.4%) | 7.749 (0%) |
| | F6 | **5.889** | 5.986 (+1.6%) | 5.691 (−3.4%) | 3.806 (−35.4%) |

| | | | | | |
|---|---|---|---|---|---|
| | F1:F6 | **5.288** | 5.187 (−1.9%) | 5.25 (−0.7%) | 2.762 (−47.8%) |
| | F1 | **5.012** | 4.725 (−5.7%) | 5.066 (+1.1%) | 1.384 (−72.4%) |
| | F6:F7 | **3.202** | 3.034 (−5.2%) | 3.009 (−6%) | 1.483 (−53.7%) |
| pumadyn32nh (n: 8192, p: 33, D: 5) | tau4:theta5 | **0.0183** | 0.01843 (+0.7%) | 0.01848 (+1%) | 0.0183 (0%) |
| | tau4 | **0.01223** | 0.01181 (−3.4%) | 0.01267 (+3.6%) | 0.01229 (+0.5%) |
| | theta5 | **0.0009914** | 0.002607 (+163%) | 0.003528 (+255.9%) | 0.001023 (+3.2%) |
| | theta3 | **0.0006446** | 0.0005413 (−16%) | 0.0005821 (−9.7%) | 0.0006446 (0%) |
| | da5 | **0.0005979** | 0.0006811 (+13.9%) | 0.0005471 (−8.5%) | 0.0005952 (−0.5%) |
| | tau1 | **0.0003974** | 0.0003712 (−6.6%) | 0.0005881 (+48%) | 0.0003911 (−1.6%) |
| | Intercept | **-0.0003658** | −0.002458 (+572%) | −0.0001511 (−58.7%) | −0.0003658 (0%) |
| red_wine (n: 1599, p: 12, D: 5) | Intercept | **5.636** | 5.675 (+0.7%) | 5.635 (0%) | 5.636 (0%) |
| | alcohol | **0.2392** | 0.2238 (−6.4%) | 0.2418 (+1.1%) | 0.2608 (+9%) |
| | sulphates | **0.1847** | 0.2005 (+8.6%) | 0.1901 (+2.9%) | 0.1821 (−1.4%) |
| | volatile_acidity | **0.1125** | 0.1326 (+17.9%) | 0.1058 (−6%) | 0.1349 (+19.9%) |
| | total_sulfur_dioxide | **0.1119** | 0.1233 (+10.2%) | 0.1221 (+9.1%) | 0.09211 (−17.7%) |
| sarcos (n: 48933, p: 22, D: 5) | V15 | **14.41** | 14.24 (−1.2%) | 14.46 (+0.3%) | 10.4 (−27.8%) |
| | Intercept | **13.66** | 14.2 (+4%) | 14.06 (+2.9%) | 13.66 (0%) |
| | V1 | **4.656** | 4.739 (+1.8%) | 4.823 (+3.6%) | 3.965 (−14.8%) |
| | V4 | **4.246** | 3.886 (−8.5%) | 4.125 (−2.8%) | 4.372 (+3%) |
| | V18 | **3.735** | 3.858 (+3.3%) | 3.556 (−4.8%) | 3.261 (−12.7%) |
| socmob (n: 1156, p: 6, D: 3) | counts_for_sons_first_occupation | **18.73** | 15.01 (−19.9%) | 19.92 (+6.4%) | 18.32 (−2.2%) |
| | Intercept | **18.21** | 10.91 (−40.1%) | 20.6 (+13.1%) | 18.21 (0%) |
| | counts_for_sons_first_occupation:sons_occupation.Manager | **3.427** | 2.298 (−32.9%) | 3.326 (−2.9%) | 3.207 (−6.4%) |
| | sons_occupation.Manager | **2.875** | 1.611 (−44%) | 3.047 (+6%) | 2.704 (−5.9%) |
| | race.white | **2.812** | 2.402 (−14.6%) | 3.054 (+8.6%) | 2.812 (0%) |
| | family_structure.nonintact | **2.621** | 2.557 (−2.4%) | 2.712 (+3.5%) | 2.544 (−2.9%) |
| | family_structure.nonintact:race.white | **2.254** | 2.618 (+16.1%) | 2.441 (+8.3%) | 2.254 (0%) |
| solar_flare (n: 1066, p: 11, D: 1) | Intercept | **0.3031** | 0.3303 (+9%) | 0.2313 (−23.7%) | 0.3031 (0%) |
| | class.E | **0.1138** | 0.1201 (+5.5%) | 0.09706 (−14.7%) | 0.1138 (0%) |
| | spot_distribution.I | **0.07133** | 0.0777 (+8.9%) | 0.05889 (−17.4%) | 0.07133 (0%) |
| | activity.2 | **0.05658** | 0.0545 (−3.7%) | 0.05299 (−6.3%) | 0.05658 (0%) |
| | largest_spot_size.K | **0.03674** | 0.04359 (+18.6%) | 0.02733 (−25.6%) | 0.03674 (0%) |
| space_ga (n: 3107, p: 7, D: 4) | houses:pop | **0.3214** | 0.318 (−1.1%) | 0.3215 (0%) | 0.1294 (−59.7%) |
| | pop | **0.3201** | 0.3209 (+0.2%) | 0.3205 (+0.1%) | 0.1387 (−56.7%) |

| | | | | | |
|---|---|---|---|---|---|
| | houses | **0.2858** | 0.3025 (+5.8%) | 0.2819 (−1.4%) | 0.1585 (−44.5%) |
| | income | **0.08183** | 0.07806 (−4.6%) | 0.07767 (−5.1%) | 0.04229 (−48.3%) |
| | ycoord | **0.07958** | 0.08196 (+3%) | 0.07674 (−3.6%) | 0.07708 (−3.1%) |
| | education | **0.07759** | 0.07654 (−1.4%) | 0.08197 (+5.6%) | 0.05881 (−24.2%) |
| | Intercept | **-0.5762** | −0.599 (+4%) | −0.5409 (−6.1%) | −0.5762 (0%) |
| student_performance_por (n: 649, p: 31, D: 3) | Intercept | **11.9** | 12.21 (+2.6%) | 11.57 (−2.8%) | 11.9 (0%) |
| | failures | **0.6817** | 0.6963 (+2.1%) | 0.7445 (+9.2%) | 0.7618 (+11.8%) |
| | school.MS | **0.4385** | 0.4249 (−3.1%) | 0.4602 (+4.9%) | 0.4582 (+4.5%) |
| | higher.yes | **0.2677** | 0.2676 (0%) | 0.2621 (−2.1%) | 0.2744 (+2.5%) |
| | goout | **0.2162** | 0.2076 (−4%) | 0.2396 (+10.8%) | 0.2112 (−2.3%) |
| | Fedu | **0.2105** | 0.195 (−7.4%) | 0.2354 (+11.8%) | 0.2159 (+2.6%) |
| | studytime | **0.2069** | 0.2611 (+26.2%) | 0.2139 (+3.4%) | 0.2157 (+4.3%) |
| superconductivity (n: 21263, p: 82, D: 5) | Intercept | **34.41** | 37.6 (+9.3%) | 37.28 (+8.3%) | 34.41 (0%) |
| | range_ThermalConductivity | **8.706** | 10.24 (+17.6%) | 9.074 (+4.2%) | 12.6 (+44.7%) |
| | range_atomic_radius | **5.834** | 6.994 (+19.9%) | 6.612 (+13.3%) | 10.07 (+72.6%) |
| | wtd_gmean_Valence | **5.257** | 5.338 (+1.5%) | 5.441 (+3.5%) | 4.771 (−9.2%) |
| | range_atomic_radius:wtd_gmean_Valence | **3.253** | 3.765 (+15.7%) | 3.044 (−6.4%) | 2.433 (−25.2%) |
| | wtd_mean_ThermalConductivity | **2.174** | 1.663 (−23.5%) | 2.248 (+3.4%) | 3.025 (+39.1%) |
| video_transcoding (n: 68784, p: 19, D: 5) | Intercept | **9.996** | 13.3 (+33.1%) | 8.713 (−12.8%) | 9.996 (0%) |
| | o_height | **7.135** | 8.856 (+24.1%) | 7.088 (−0.7%) | 7.133 (0%) |
| | o_codec.h264 | **6.294** | 9.064 (+44%) | 5.311 (−15.6%) | 5.611 (−10.9%) |
| | o_codec.h264:o_height | **5.193** | 6.577 (+26.7%) | 4.903 (−5.6%) | 4.649 (−10.5%) |
| | o_bitrate:o_codec.h264 | **2.71** | 3.967 (+46.4%) | 2.417 (−10.8%) | 2.647 (−2.3%) |
| wave_energy (n: 72000, p: 49, D: 4) | Intercept | **3760000** | 3754000 (−0.2%) | 3740000 (−0.5%) | 3760000 (0%) |
| | energy4 | **23350** | 25030 (+7.2%) | 23670 (+1.4%) | 23620 (+1.2%) |
| | energy3 | **23100** | 22790 (−1.3%) | 22490 (−2.6%) | 22170 (−4%) |
| | energy9 | **23030** | 23490 (+2%) | 23520 (+2.1%) | 22100 (−4%) |
| | energy7 | **22980** | 23420 (+1.9%) | 23150 (+0.7%) | 22150 (−3.6%) |
| | energy12 | **22730** | 22310 (−1.8%) | 23030 (+1.3%) | 22170 (−2.5%) |
| | energy15 | **22650** | 22340 (−1.4%) | 22700 (+0.2%) | 22640 (0%) |
| | energy10 | **22640** | 22440 (−0.9%) | 21820 (−3.6%) | 22570 (−0.3%) |
| | energy14 | **22500** | 22620 (+0.5%) | 23320 (+3.6%) | 21270 (−5.5%) |
| | energy5 | **22370** | 23210 (+3.8%) | 23710 (+6%) | 22060 (−1.4%) |
| white_wine (n: 4898, p: 12, D: 5) | Intercept | **5.878** | 5.844 (−0.6%) | 5.89 (+0.2%) | 5.878 (0%) |

| | | | | |
|---|---|---|---|---|
| alcohol | **0.2482** | 0.2618 (+5.5%) | 0.2396 (−3.5%) | 0.2358 (−5%) |
| density | **0.1564** | 0.1409 (−9.9%) | 0.1628 (+4.1%) | 0.1123 (−28.2%) |
| residual_sugar | **0.1461** | 0.1617 (+10.7%) | 0.1613 (+10.4%) | 0.1179 (−19.3%) |
| volatile_acidity | **0.1275** | 0.1271 (−0.3%) | 0.1218 (−4.5%) | 0.1296 (+1.6%) |
| free_sulfur_dioxide | **0.1137** | 0.1274 (+12%) | 0.1416 (+24.5%) | 0.113 (−0.6%) |

### D.3. FastPD vs Path-dependent (Classification)

For the classification experiments, we utilized the OpenML-CC18 Task Collection. The experimental procedure, including 5-fold cross-validation and XGBoost hyperparameter tuning, mirrored that of the regression setting. However, datasets with over 500 features were excluded since they primarily were image-based datasets.

For multiclass problems (where the number of classes $K > 2$), the default approach of XGBoost involves training $K$ separate binary classification models ($\hat{f}_k$), each outputting a raw score and distinguishing one class from the rest (one-vs-rest). The interpretability methods were then applied to the sum of these raw scores, $\hat{f} = \sum_{k=1}^{K} \hat{f}_k$.

| Dataset | Variable/Interaction | FastPD (Reference) | FastPD (50) | FastPD (100) | Path-dependent |
|---|---|---|---|---|---|
| GesturePhaseSegmentationProcessed (n: 9873, p: 33, D: 5) | X26 | **0.5959** | 0.6367 (+6.8% ) | 0.616 (+3.4% ) | 0.5459 (−8.4%) |
| | X28 | **0.5427** | 0.6583 (+21.3% ) | 0.5467 (+0.7% ) | 0.5737 (+5.7%) |
| | X5 | **0.3939** | 0.5222 (+32.6% ) | 0.5049 (+28.2% ) | 0.3717 (−5.6%) |
| | X25 | **0.371** | 0.4044 (+9% ) | 0.324 (−12.7% ) | 0.324 (−12.7%) |
| | X11 | **0.3238** | 0.3388 (+4.6% ) | 0.3537 (+9.2% ) | 0.3915 (+20.9%) |
| | Intercept | **-3.879** | −3.606 (−7% ) | −4.097 (+5.6% ) | −3.879 (0%) |
| MiceProtein (n: 1080, p: 78, D: 2) | SOD1_N | **1.752** | 1.784 (+1.8% ) | 1.792 (+2.3% ) | 1.875 (+7%) |
| | pCAMKII_N | **1.226** | 1.307 (+6.6% ) | 1.303 (+6.3% ) | 1.196 (−2.4%) |
| | GluR3_N | **1.059** | 1.062 (+0.3% ) | 1.05 (−0.8% ) | 0.9862 (−6.9%) |
| | BRAF_N | **1.057** | 1.121 (+6.1% ) | 1.06 (+0.3% ) | 1.047 (−0.9%) |
| | CaNA_N | **0.9413** | 1.015 (+7.8% ) | 0.9779 (+3.9% ) | 1.031 (+9.5%) |
| | Intercept | **-25.18** | −24.91 (−1.1% ) | −24.92 (−1% ) | −25.18 (0%) |
| PhishingWebsites (n: 11055, p: 31, D: 5) | URL_of_Anchor.0 | **3.328** | 3.581 (+7.6% ) | 3.313 (−0.5% ) | 3.187 (−4.2%) |
| | SSLfinal_State.1 | **3.044** | 3.45 (+13.3% ) | 3.026 (−0.6% ) | 3.991 (+31.1%) |
| | URL_of_Anchor.1 | **2.665** | 2.351 (−11.8% ) | 2.615 (−1.9% ) | 2.373 (−11%) |
| | URL_of_Anchor.0:URL_of_Anchor.1 | **1.646** | 1.353 (−17.8% ) | 1.705 (+3.6% ) | 1.642 (−0.2%) |
| | Intercept | **0.07477** | 0.4047 (+441.3% ) | −0.9864 (−1419.2% ) | 0.07477 (0%) |
| adult (n: 48842, p: 15, D: 5) | age | **0.7777** | 0.8597 (+10.5% ) | 0.7766 (−0.1% ) | 0.8176 (+5.1%) |
| | capital.gain | **0.6253** | 0.4404 (−29.6% ) | 0.5037 (−19.4% ) | 0.6606 (+5.6%) |
| | marital.status.Never.married | **0.6014** | 0.6082 (+1.1% ) | 0.6356 (+5.7% ) | 0.6055 (+0.7%) |
| | education.num | **0.3952** | 0.3975 (+0.6% ) | 0.3989 (+0.9% ) | 0.4742 (+20%) |
| | Intercept | **-2.074** | −2.7 (+30.2% ) | −2.572 (+24% ) | −2.074 (0%) |
| analcatdata_authorship (n: 841, p: 71, D: 1) | the | **1.519** | 1.514 (−0.3% ) | 1.525 (+0.4% ) | 1.519 (0%) |
| | was | **1.326** | 1.37 (+3.3% ) | 1.325 (−0.1% ) | 1.326 (0%) |
| | her | **1.1** | 1.102 (+0.2% ) | 1.11 (+0.9% ) | 1.1 (0%) |
| | should | **0.8523** | 0.8532 (+0.1% ) | 0.837 (−1.8% ) | 0.8523 (0%) |
| | Intercept | **-8.48** | −8.531 (+0.6% ) | −8.648 (+2% ) | −8.48 (0%) |
| analcatdata_dmft (n: 797, p: 5, D: 1) | Intercept | **0.4323** | 0.5592 (+29.4% ) | 0.3711 (−14.2% ) | 0.4323 (0%) |
| | Ethnic.White | **0.2147** | 0.2171 (+1.1% ) | 0.2132 (−0.7% ) | 0.2147 (0%) |
| | Ethnic.Dark | **0.119** | 0.1117 (−6.1% ) | 0.1234 (+3.7% ) | 0.119 (0%) |
| | DMFT.Begin.4 | **0.03935** | 0.05373 (+36.5% ) | 0.03803 (−3.4% ) | 0.03935 (0%) |
| | DMFT.End.5 | **0.03664** | 0.03443 (−6% ) | 0.0418 (+14.1% ) | 0.03664 (0%) |
| balance-scale (n: 625, p: 5, D: 2) | left.distance | **0.9896** | 1.081 (+9.2% ) | 1.005 (+1.6% ) | 0.9896 (0%) |

| | | | | | |
|---|---|---|---|---|---|
| | right.weight | **0.5251** | 0.4571 (−12.9% ) | 0.55 (+4.7% ) | 0.5251 (0%) |
| | right.distance:right.weight | **0.4602** | 0.4382 (−4.8% ) | 0.4897 (+6.4% ) | 0.4602 (0%) |
| | left.distance:right.distance | **0.441** | 0.4379 (−0.7% ) | 0.4528 (+2.7% ) | 0.441 (0%) |
| | Intercept | **-2.739** | −2.546 (−7% ) | −2.742 (+0.1% ) | −2.739 (0%) |
| bank-marketing (n: 45211, p: 17, D: 5) | Intercept | **4.181** | 4.418 (+5.7% ) | 4.298 (+2.8% ) | 4.181 (0%) |
| | V12 | **1.257** | 1.272 (+1.2% ) | 1.232 (−2% ) | 1.287 (+2.4%) |
| | V9.unknown | **0.5507** | 0.3857 (−30% ) | 0.669 (+21.5% ) | 0.6991 (+26.9%) |
| | V11.may | **0.3244** | 0.3225 (−0.6% ) | 0.3487 (+7.5% ) | 0.2573 (−20.7%) |
| | V12:V9.unknown | **0.2655** | 0.2475 (−6.8% ) | 0.2858 (+7.6% ) | 0.308 (+16%) |
| | V10 | **0.2515** | 0.2542 (+1.1% ) | 0.169 (−32.8% ) | 0.1644 (−34.6%) |
| banknote-authentication (n: 1372, p: 5, D: 2) | V1 | **5.331** | 5.392 (+1.1% ) | 5.242 (−1.7% ) | 5.306 (−0.5%) |
| | V2 | **4.571** | 4.732 (+3.5% ) | 4.696 (+2.7% ) | 4.399 (−3.8%) |
| | V3 | **4.31** | 4.305 (−0.1% ) | 4.218 (−2.1% ) | 3.78 (−12.3%) |
| | Intercept | **2.143** | 1.427 (−33.4% ) | 3.174 (+48.1% ) | 2.143 (0%) |
| | V2:V3 | **0.8987** | 0.8774 (−2.4% ) | 0.8531 (−5.1% ) | 0.8824 (−1.8%) |
| blood-transfusion-service-center (n: 748, p: 5, D: 1) | Intercept | **1.907** | 2.078 (+9% ) | 1.9 (−0.4% ) | 1.907 (0%) |
| | V1 | **0.6442** | 0.6441 (0% ) | 0.6447 (+0.1% ) | 0.6442 (0%) |
| | V2 | **0.4394** | 0.4455 (+1.4% ) | 0.4422 (+0.6% ) | 0.4394 (0%) |
| | V4 | **0.3667** | 0.4028 (+9.8% ) | 0.3913 (+6.7% ) | 0.3667 (0%) |
| | V3 | **0.156** | 0.1566 (+0.4% ) | 0.1576 (+1% ) | 0.156 (0%) |
| breast-w (n: 699, p: 10, D: 1) | Intercept | **2.177** | 1.714 (−21.3% ) | 1.977 (−9.2% ) | 2.177 (0%) |
| | Bare_Nuclei | **1.122** | 1.147 (+2.2% ) | 1.139 (+1.5% ) | 1.122 (0%) |
| | Cell_Size_Uniformity | **0.879** | 0.9123 (+3.8% ) | 0.8931 (+1.6% ) | 0.879 (0%) |
| | Clump_Thickness | **0.6338** | 0.5865 (−7.5% ) | 0.6205 (−2.1% ) | 0.6338 (0%) |
| | Cell_Shape_Uniformity | **0.6229** | 0.6393 (+2.6% ) | 0.627 (+0.7% ) | 0.6229 (0%) |
| | Bland_Chromatin | **0.5479** | 0.5897 (+7.6% ) | 0.5402 (−1.4% ) | 0.5479 (0%) |
| car (n: 1728, p: 7, D: 5) | lug_boot.big | **1.925** | 1.703 (−11.5% ) | 1.762 (−8.5% ) | 1.741 (−9.6%) |
| | buying.low | **1.324** | 1.01 (−23.7% ) | 1.259 (−4.9% ) | 1.324 (0%) |
| | maint.low | **1.282** | 1.389 (+8.3% ) | 1.156 (−9.8% ) | 1.253 (−2.3%) |
| | buying.med | **1.274** | 1.551 (+21.7% ) | 1.415 (+11.1% ) | 1.364 (+7.1%) |
| | maint.med | **1.097** | 1.179 (+7.5% ) | 1.15 (+4.8% ) | 1.041 (−5.1%) |
| | doors.5more | **1.029** | 1.051 (+2.1% ) | 1.16 (+12.7% ) | 0.9159 (−11%) |
| | Intercept | **-10.95** | −10.85 (−0.9% ) | −11.23 (+2.6% ) | −10.95 (0%) |
| churn (n: 5000, p: 21, D: 5) | Intercept | **3.316** | 3.108 (−6.3% ) | 3.528 (+6.4% ) | 3.316 (0%) |

| | | | | | |
|---|---|---|---|---|---|
| | total_day_charge | **0.4751** | 0.5453 (+14.8% ) | 0.3872 (−18.5% ) | 0.4808 (+1.2%) |
| | international_plan.1 | **0.3634** | 0.3857 (+6.1% ) | 0.314 (−13.6% ) | 0.3752 (+3.2%) |
| | international_plan.1:total_intl_calls | **0.2522** | 0.3301 (+30.9% ) | 0.2674 (+6% ) | 0.2373 (−5.9%) |
| | number_customer_service_calls.4 | **0.226** | 0.2262 (+0.1% ) | 0.1725 (−23.7% ) | 0.2172 (−3.9%) |
| | total_intl_calls | **0.1968** | 0.2782 (+41.4% ) | 0.2155 (+9.5% ) | 0.1921 (−2.4%) |
| | number_customer_service_calls.5 | **0.1169** | 0.2977 (+154.7% ) | 0.1737 (+48.6% ) | 0.1203 (+2.9%) |
| climate-model-simulation-crashes (n: 540, p: 19, D: 2) | vconst_corr | **1.001** | 0.8937 (−10.7% ) | 1.028 (+2.7% ) | 1.002 (+0.1%) |
| | vconst_2 | **0.9663** | 0.9107 (−5.8% ) | 1.023 (+5.9% ) | 0.9713 (+0.5%) |
| | convect_corr | **0.7112** | 0.6609 (−7.1% ) | 0.7399 (+4% ) | 0.7023 (−1.3%) |
| | bckgrnd_vdc1 | **0.4785** | 0.4181 (−12.6% ) | 0.4985 (+4.2% ) | 0.4794 (+0.2%) |
| | vconst_2:vconst_corr | **0.4772** | 0.465 (−2.6% ) | 0.4816 (+0.9% ) | 0.4773 (0%) |
| | Intercept | **-3.572** | −3.997 (+11.9% ) | −3.505 (−1.9% ) | −3.572 (0%) |
| cmc (n: 1473, p: 10, D: 3) | Intercept | **0.2812** | 0.311 (+10.6% ) | 0.2435 (−13.4% ) | 0.2812 (0%) |
| | Number_of_children_ever_born | **0.1651** | 0.1626 (−1.5% ) | 0.1678 (+1.6% ) | 0.1881 (+13.9%) |
| | Wifes_education.4 | **0.1497** | 0.1467 (−2% ) | 0.1324 (−11.6% ) | 0.135 (−9.8%) |
| | Wifes_education.3 | **0.1123** | 0.1067 (−5% ) | 0.1117 (−0.5% ) | 0.1051 (−6.4%) |
| | Wifes_age | **0.1016** | 0.1022 (+0.6% ) | 0.1076 (+5.9% ) | 0.08719 (−14.2%) |
| connect-4 (n: 67557, p: 43, D: 5) | g1.1 | **0.3457** | 0.405 (+17.2% ) | 0.3575 (+3.4% ) | 0.3156 (−8.7%) |
| | a1.1 | **0.3032** | 0.2637 (−13% ) | 0.2406 (−20.6% ) | 0.2835 (−6.5%) |
| | c2.2 | **0.1752** | 0.1971 (+12.5% ) | 0.161 (−8.1% ) | 0.1605 (−8.4%) |
| | d1.2 | **0.1502** | 0.1682 (+12% ) | 0.1451 (−3.4% ) | 0.1834 (+22.1%) |
| | d2.1 | **0.1377** | 0.1346 (−2.3% ) | 0.1748 (+26.9% ) | 0.1006 (−26.9%) |
| | Intercept | **-0.6306** | −0.7717 (+22.4% ) | −0.3145 (−50.1% ) | −0.6306 (0%) |
| credit-approval (n: 690, p: 16, D: 2) | A9.f | **1.606** | 1.64 (+2.1% ) | 1.62 (+0.9% ) | 1.71 (+6.5%) |
| | A15 | **0.489** | 0.4922 (+0.7% ) | 0.5052 (+3.3% ) | 0.5344 (+9.3%) |
| | A8 | **0.338** | 0.3257 (−3.6% ) | 0.3255 (−3.7% ) | 0.351 (+3.8%) |
| | A11 | **0.2694** | 0.2514 (−6.7% ) | 0.2939 (+9.1% ) | 0.3002 (+11.4%) |
| | Intercept | **0.1466** | 0.1272 (−13.2% ) | −0.02637 (−118% ) | 0.1466 (0%) |
| credit-g (n: 1000, p: 21, D: 4) | Intercept | **1.708** | 1.481 (−13.3% ) | 1.744 (+2.1% ) | 1.708 (0%) |
| | checking_status.no.checking | **0.6726** | 0.7092 (+5.4% ) | 0.6742 (+0.2% ) | 0.7091 (+5.4%) |
| | duration | **0.3661** | 0.3567 (−2.6% ) | 0.3834 (+4.7% ) | 0.3751 (+2.5%) |
| | credit_amount | **0.3079** | 0.3182 (+3.3% ) | 0.3174 (+3.1% ) | 0.2982 (−3.2%) |
| | credit_history.critical.other.existing.credit | **0.2568** | 0.2282 (−11.1% ) | 0.2725 (+6.1% ) | 0.2447 (−4.7%) |
| | credit_amount:duration | **0.2043** | 0.251 (+22.9% ) | 0.2353 (+15.2% ) | 0.1479 (−27.6%) |

| | | | | | |
|---|---|---|---|---|---|
| cylinder-bands (n: 540, p: 38, D: 5) | press_speed | **0.4623** | 0.4539 (−1.8% ) | 0.4728 (+2.3% ) | 0.4688 (+1.4%) |
| | ink_pct | **0.3341** | 0.3537 (+5.9% ) | 0.3564 (+6.7% ) | 0.3298 (−1.3%) |
| | ESA_Voltage | **0.3306** | 0.3192 (−3.4% ) | 0.3274 (−1% ) | 0.3402 (+2.9%) |
| | press_type.woodhoe70 | **0.2944** | 0.2937 (−0.2% ) | 0.3007 (+2.1% ) | 0.2755 (−6.4%) |
| | solvent_pct | **0.2894** | 0.2708 (−6.4% ) | 0.2735 (−5.5% ) | 0.285 (−1.5%) |
| | Intercept | **-0.4583** | −0.0698 (−84.8% ) | −0.3707 (−19.1% ) | −0.4583 (0%) |
| diabetes (n: 768, p: 9, D: 1) | Intercept | **1.378** | 1.536 (+11.5% ) | 1.231 (−10.7% ) | 1.378 (0%) |
| | plas | **0.7336** | 0.7202 (−1.8% ) | 0.7339 (0% ) | 0.7336 (0%) |
| | mass | **0.5288** | 0.5546 (+4.9% ) | 0.5056 (−4.4% ) | 0.5288 (0%) |
| | age | **0.3541** | 0.3525 (−0.5% ) | 0.352 (−0.6% ) | 0.3541 (0%) |
| | pedi | **0.1807** | 0.1833 (+1.4% ) | 0.1787 (−1.1% ) | 0.1807 (0%) |
| dna (n: 3186, p: 181, D: 5) | A92 | **0.9427** | 0.9694 (+2.8% ) | 0.9219 (−2.2% ) | 1.137 (+20.6%) |
| | A84 | **0.51** | 0.4448 (−12.8% ) | 0.5111 (+0.2% ) | 0.6598 (+29.4%) |
| | A89 | **0.488** | 0.4886 (+0.1% ) | 0.5167 (+5.9% ) | 0.8322 (+70.5%) |
| | A93 | **0.4782** | 0.4644 (−2.9% ) | 0.3591 (−24.9% ) | 0.4856 (+1.5%) |
| | A95 | **0.4725** | 0.4878 (+3.2% ) | 0.4933 (+4.4% ) | 0.4948 (+4.7%) |
| | A94 | **0.4435** | 0.4833 (+9% ) | 0.447 (+0.8% ) | 0.453 (+2.1%) |
| | Intercept | **-3.09** | −2.91 (−5.8% ) | −3.245 (+5% ) | −3.09 (0%) |
| dresses-sales (n: 500, p: 13, D: 1) | Intercept | **0.7038** | 0.8286 (+17.7% ) | 0.7303 (+3.8% ) | 0.7038 (0%) |
| | V6.Spring | **0.2741** | 0.2489 (−9.2% ) | 0.2785 (+1.6% ) | 0.2741 (0%) |
| | V10.rayon | **0.1774** | 0.1663 (−6.3% ) | 0.1842 (+3.8% ) | 0.1774 (0%) |
| | V10.cotton | **0.1542** | 0.1557 (+1% ) | 0.1546 (+0.3% ) | 0.1542 (0%) |
| | V8.short | **0.126** | 0.1367 (+8.5% ) | 0.1246 (−1.1% ) | 0.126 (0%) |
| electricity (n: 45312, p: 9, D: 5) | nswprice | **3.821** | 3.881 (+1.6% ) | 3.923 (+2.7% ) | 3.188 (−16.6%) |
| | date | **2.297** | 2.688 (+17% ) | 2.224 (−3.2% ) | 1.796 (−21.8%) |
| | date:nswprice | **1.55** | 1.407 (−9.2% ) | 1.503 (−3% ) | 1.327 (−14.4%) |
| | nswprice:vicprice | **0.3858** | 0.6725 (+74.3% ) | 0.4202 (+8.9% ) | 0.3269 (−15.3%) |
| | vicprice | **0.3618** | 0.4327 (+19.6% ) | 0.4099 (+13.3% ) | 0.4026 (+11.3%) |
| | Intercept | **-0.5675** | −1.567 (+176.1% ) | −0.5729 (+1% ) | −0.5675 (0%) |
| eucalyptus (n: 736, p: 20, D: 2) | Vig | **0.6912** | 0.6582 (−4.8% ) | 0.6811 (−1.5% ) | 0.7452 (+7.8%) |
| | Ht | **0.5555** | 0.5429 (−2.3% ) | 0.5752 (+3.5% ) | 0.5935 (+6.8%) |
| | Crown_Fm | **0.423** | 0.4139 (−2.2% ) | 0.44 (+4% ) | 0.4552 (+7.6%) |
| | Surv | **0.3366** | 0.3796 (+12.8% ) | 0.3976 (+18.1% ) | 0.3338 (−0.8%) |
| | Intercept | **-4.564** | −4.236 (−7.2% ) | −4.295 (−5.9% ) | −4.564 (0%) |

| | | | | | |
|---|---|---|---|---|---|
| first-order-theorem-proving (n: 6118, p: 52, D: 5) | V19 | **0.3181** | 0.2757 (−13.3% ) | 0.3417 (+7.4% ) | 0.2213 (−30.4%) |
| | V23 | **0.2871** | 0.2907 (+1.3% ) | 0.2853 (−0.6% ) | 0.3112 (+8.4%) |
| | V38 | **0.2435** | 0.2327 (−4.4% ) | 0.221 (−9.2% ) | 0.1969 (−19.1%) |
| | V25 | **0.2168** | 0.2039 (−6% ) | 0.216 (−0.4% ) | 0.1875 (−13.5%) |
| | V39 | **0.1941** | 0.2417 (+24.5% ) | 0.1892 (−2.5% ) | 0.08547 (−56%) |
| | V4 | **0.1925** | 0.2139 (+11.1% ) | 0.2219 (+15.3% ) | 0.121 (−37.1%) |
| | V11 | **0.1589** | 0.1181 (−25.7% ) | 0.2226 (+40.1% ) | 0.1219 (−23.3%) |
| | Intercept | **-4.776** | −4.622 (−3.2% ) | −4.835 (+1.2% ) | −4.776 (0%) |
| ilpd (n: 583, p: 11, D: 1) | Intercept | **1.609** | 1.48 (−8% ) | 1.514 (−5.9% ) | 1.609 (0%) |
| | V5 | **0.2843** | 0.2826 (−0.6% ) | 0.2842 (0% ) | 0.2843 (0%) |
| | V4 | **0.2712** | 0.2474 (−8.8% ) | 0.2567 (−5.3% ) | 0.2712 (0%) |
| | V6 | **0.2215** | 0.2427 (+9.6% ) | 0.2218 (+0.1% ) | 0.2215 (0%) |
| | V3 | **0.1898** | 0.1787 (−5.8% ) | 0.1847 (−2.7% ) | 0.1898 (0%) |
| jm1 (n: 10885, p: 22, D: 4) | Intercept | **2.17** | 2.035 (−6.2% ) | 2.217 (+2.2% ) | 2.17 (0%) |
| | lOBlank | **0.3751** | 0.3452 (−8% ) | 0.3802 (+1.4% ) | 0.3056 (−18.5%) |
| | loc | **0.3712** | 0.3421 (−7.8% ) | 0.3587 (−3.4% ) | 0.4542 (+22.4%) |
| | lOBlank:total_Op | **0.1968** | 0.1741 (−11.5% ) | 0.201 (+2.1% ) | 0.07209 (−63.4%) |
| | total_Op | **0.188** | 0.2046 (+8.8% ) | 0.1904 (+1.3% ) | 0.1033 (−45.1%) |
| | total_Opnd | **0.1595** | 0.1797 (+12.7% ) | 0.1506 (−5.6% ) | 0.1035 (−35.1%) |
| | lOBlank:loc | **0.1185** | 0.1251 (+5.6% ) | 0.118 (−0.4% ) | 0.1152 (−2.8%) |
| jungle_chess_2pcs_raw_endgame_complete (n: 44819, p: 7, D: 5) | white_piece0_rank | **2.148** | 2.407 (+12.1% ) | 2.14 (−0.4% ) | 2.18 (+1.5%) |
| | black_piece0_rank | **1.811** | 2.105 (+16.2% ) | 1.64 (−9.4% ) | 1.835 (+1.3%) |
| | black_piece0_rank:black_piece0_strength | **0.9972** | 1.041 (+4.4% ) | 0.9096 (−8.8% ) | 0.9841 (−1.3%) |
| | black_piece0_rank:white_piece0_strength | **0.7611** | 0.7054 (−7.3% ) | 0.782 (+2.7% ) | 0.776 (+2%) |
| | black_piece0_strength:white_piece0_strength | **0.7242** | 0.8054 (+11.2% ) | 0.789 (+8.9% ) | 0.7158 (−1.2%) |
| | Intercept | **-4.976** | −4.464 (−10.3% ) | −4.663 (−6.3% ) | −4.976 (0%) |
| kc1 (n: 2109, p: 22, D: 4) | Intercept | **2.664** | 2.423 (−9% ) | 2.599 (−2.4% ) | 2.664 (0%) |
| | lOCode | **0.3439** | 0.2703 (−21.4% ) | 0.3193 (−7.2% ) | 0.3206 (−6.8%) |
| | e | **0.2808** | 0.2559 (−8.9% ) | 0.2767 (−1.5% ) | 0.2074 (−26.1%) |
| | loc | **0.2543** | 0.2248 (−11.6% ) | 0.2558 (+0.6% ) | 0.2418 (−4.9%) |
| | i | **0.1886** | 0.149 (−21% ) | 0.1968 (+4.3% ) | 0.2447 (+29.7%) |
| | d | **0.1677** | 0.1581 (−5.7% ) | 0.1462 (−12.8% ) | 0.1092 (−34.9%) |
| kc2 (n: 522, p: 22, D: 2) | Intercept | **2.356** | 2.195 (−6.8% ) | 2.279 (−3.3% ) | 2.356 (0%) |
| | i | **0.4001** | 0.399 (−0.3% ) | 0.3871 (−3.2% ) | 0.2755 (−31.1%) |

| | | | | | |
|---|---|---|---|---|---|
| | uniq_Opnd | **0.3854** | 0.3804 (−1.3% ) | 0.374 (−3% ) | 0.3627 (−5.9%) |
| | total_Opnd | **0.3419** | 0.3641 (+6.5% ) | 0.3712 (+8.6% ) | 0.3247 (−5%) |
| | lOBlank | **0.1861** | 0.1726 (−7.3% ) | 0.2096 (+12.6% ) | 0.1824 (−2%) |
| | uniq_Op | **0.1585** | 0.1759 (+11% ) | 0.1678 (+5.9% ) | 0.1498 (−5.5%) |
| kr-vs-kp (n: 3196, p: 37, D: 5) | bxqsq.f | **3.697** | 3.539 (−4.3% ) | 3.653 (−1.2% ) | 3.647 (−1.4%) |
| | wknck.f | **3.488** | 3.915 (+12.2% ) | 3.624 (+3.9% ) | 3.346 (−4.1%) |
| | rimmx.f | **3.124** | 2.326 (−25.5% ) | 2.886 (−7.6% ) | 3.055 (−2.2%) |
| | bkxbq.f | **1.343** | 1.837 (+36.8% ) | 1.409 (+4.9% ) | 1.371 (+2.1%) |
| | Intercept | **0.7984** | 0.8956 (+12.2% ) | 0.2632 (−67% ) | 0.7984 (0%) |
| letter (n: 20000, p: 17, D: 5) | x.ege | **4.61** | 4.21 (−8.7% ) | 4.527 (−1.8% ) | 4.107 (−10.9%) |
| | y.ege | **3.441** | 3.427 (−0.4% ) | 2.935 (−14.7% ) | 3.13 (−9%) |
| | xegvy | **2.67** | 2.589 (−3% ) | 2.68 (+0.4% ) | 2.408 (−9.8%) |
| | xybar | **2.45** | 2.085 (−14.9% ) | 2.143 (−12.5% ) | 2.3 (−6.1%) |
| | x2ybr | **2.286** | 2.37 (+3.7% ) | 2.258 (−1.2% ) | 1.993 (−12.8%) |
| | Intercept | **-131.2** | −128.6 (−2% ) | −128.2 (−2.3% ) | −131.2 (0%) |
| mfeat-factors (n: 2000, p: 217, D: 2) | att1 | **1.213** | 1.191 (−1.8% ) | 1.187 (−2.1% ) | 1.222 (+0.7%) |
| | att181 | **0.9014** | 0.952 (+5.6% ) | 0.9538 (+5.8% ) | 0.9899 (+9.8%) |
| | att37 | **0.7361** | 0.8493 (+15.4% ) | 0.8563 (+16.3% ) | 0.9363 (+27.2%) |
| | att194 | **0.6411** | 0.6299 (−1.7% ) | 0.6215 (−3.1% ) | 0.6047 (−5.7%) |
| | att213 | **0.6099** | 0.6938 (+13.8% ) | 0.6682 (+9.6% ) | 0.6421 (+5.3%) |
| | Intercept | **-37.82** | −38.57 (+2% ) | −37.7 (−0.3% ) | −37.82 (0%) |
| mfeat-fourier (n: 2000, p: 77, D: 5) | att7 | **0.9039** | 0.8597 (−4.9% ) | 0.805 (−10.9% ) | 1.171 (+29.5%) |
| | att73 | **0.8344** | 0.8666 (+3.9% ) | 0.891 (+6.8% ) | 1.116 (+33.7%) |
| | att2 | **0.8336** | 0.8625 (+3.5% ) | 0.7984 (−4.2% ) | 0.7765 (−6.8%) |
| | att76 | **0.5851** | 0.4715 (−19.4% ) | 0.5686 (−2.8% ) | 0.6125 (+4.7%) |
| | att6 | **0.54** | 0.601 (+11.3% ) | 0.6122 (+13.4% ) | 0.6534 (+21%) |
| | Intercept | **-26.19** | −26.55 (+1.4% ) | −26.44 (+1% ) | −26.19 (0%) |
| mfeat-karhunen (n: 2000, p: 65, D: 5) | att1 | **1.469** | 1.548 (+5.4% ) | 1.439 (−2% ) | 1.522 (+3.6%) |
| | att3 | **1.318** | 1.246 (−5.5% ) | 1.301 (−1.3% ) | 1.487 (+12.8%) |
| | att4 | **1.126** | 1.338 (+18.8% ) | 1.064 (−5.5% ) | 1.133 (+0.6%) |
| | att7 | **0.9811** | 1.109 (+13% ) | 0.9565 (−2.5% ) | 1.126 (+14.8%) |
| | att2 | **0.8767** | 0.7276 (−17% ) | 1.049 (+19.7% ) | 0.9546 (+8.9%) |
| | Intercept | **-33.11** | −33.67 (+1.7% ) | −33.2 (+0.3% ) | −33.11 (0%) |
| mfeat-morphological (n: 2000, p: 7, D: 1) | att4 | **3.748** | 3.794 (+1.2% ) | 3.589 (−4.2% ) | 3.748 (0%) |

| | | | | | |
|---|---|---|---|---|---|
| | att2 | **3.318** | 3.212 (−3.2%) | 3.308 (−0.3%) | 3.318 (0%) |
| | att1 | **2.913** | 2.857 (−1.9%) | 2.921 (+0.3%) | 2.913 (0%) |
| | att6 | **1.96** | 1.964 (+0.2%) | 1.957 (−0.2%) | 1.96 (0%) |
| | Intercept | **-21.72** | −23.36 (+7.6%) | −22.37 (+3%) | −21.72 (0%) |
| mfeat-pixel (n: 2000, p: 241, D: 5) | att162 | **0.4647** | 0.4508 (−3%) | 0.4105 (−11.7%) | 0.6221 (+33.9%) |
| | att19 | **0.4553** | 0.3895 (−14.5%) | 0.4393 (−3.5%) | 0.507 (+11.4%) |
| | att57 | **0.4189** | 0.4877 (+16.4%) | 0.4943 (+18%) | 0.4012 (−4.2%) |
| | att214 | **0.412** | 0.4123 (+0.1%) | 0.4044 (−1.8%) | 0.4574 (+11%) |
| | att153 | **0.3907** | 0.4174 (+6.8%) | 0.442 (+13.1%) | 0.585 (+49.7%) |
| | att113 | **0.3706** | 0.4066 (+9.7%) | 0.359 (−3.1%) | 0.5165 (+39.4%) |
| | Intercept | **-31.93** | −31.29 (−2%) | −31.89 (−0.1%) | −31.93 (0%) |
| mfeat-zernike (n: 2000, p: 48, D: 2) | att29 | **1.137** | 1.151 (+1.2%) | 1.141 (+0.4%) | 1.186 (+4.3%) |
| | att36 | **1.029** | 1.11 (+7.9%) | 1.068 (+3.8%) | 1.028 (−0.1%) |
| | att45 | **0.9863** | 0.9713 (−1.5%) | 0.9926 (+0.6%) | 0.7626 (−22.7%) |
| | att43 | **0.958** | 1.018 (+6.3%) | 0.9376 (−2.1%) | 0.9981 (+4.2%) |
| | att33 | **0.9426** | 0.9923 (+5.3%) | 0.9413 (−0.1%) | 0.8801 (−6.6%) |
| | att19 | **0.8309** | 0.8354 (+0.5%) | 0.7189 (−13.5%) | 1.071 (+28.9%) |
| | Intercept | **-26.83** | −26.25 (−2.2%) | −27.78 (+3.5%) | −26.83 (0%) |
| nomao (n: 34465, p: 119, D: 5) | V6 | **1.002** | 0.9554 (−4.7%) | 1.065 (+6.3%) | 1.403 (+40%) |
| | V90 | **0.8547** | 0.7562 (−11.5%) | 0.8676 (+1.5%) | 0.9458 (+10.7%) |
| | V4 | **0.6198** | 0.6085 (−1.8%) | 0.6263 (+1%) | 0.676 (+9.1%) |
| | V100.3 | **0.5639** | 0.5918 (+4.9%) | 0.6077 (+7.8%) | 0.8928 (+58.3%) |
| | V61 | **0.5574** | 0.593 (+6.4%) | 0.6077 (+9%) | 0.4243 (−23.9%) |
| | V1 | **0.5561** | 0.5659 (+1.8%) | 0.6561 (+18%) | 0.8947 (+60.9%) |
| | Intercept | **-3.063** | −2.635 (−14%) | −1.931 (−37%) | −3.063 (0%) |
| numerai28.6 (n: 96320, p: 22, D: 2) | Intercept | **0.4794** | 0.4686 (−2.3%) | 0.4916 (+2.5%) | 0.4794 (0%) |
| | attribute_13 | **0.03296** | 0.03171 (−3.8%) | 0.03317 (+0.6%) | 0.03048 (−7.5%) |
| | attribute_1 | **0.02981** | 0.02981 (0%) | 0.02936 (−1.5%) | 0.02959 (−0.7%) |
| | attribute_14 | **0.02838** | 0.02969 (+4.6%) | 0.02848 (+0.4%) | 0.02714 (−4.4%) |
| | attribute_16 | **0.0274** | 0.02492 (−9.1%) | 0.02681 (−2.2%) | 0.0282 (+2.9%) |
| optdigits (n: 5620, p: 65, D: 4) | input27 | **1.119** | 1.15 (+2.8%) | 1.181 (+5.5%) | 1.396 (+24.8%) |
| | input63 | **0.8117** | 0.8321 (+2.5%) | 0.7816 (−3.7%) | 0.9487 (+16.9%) |
| | input37 | **0.7809** | 0.8437 (+8%) | 0.8531 (+9.2%) | 1.058 (+35.5%) |
| | input54 | **0.7408** | 0.8106 (+9.4%) | 0.7068 (−4.6%) | 0.8397 (+13.4%) |

| Dataset | Variable | True | Est 1 | Est 2 | Est 3 |
|---|---|---|---|---|---|
| | Intercept | **-35.07** | −34.46 (−1.7% ) | −34.8 (−0.8% ) | −35.07 (0%) |
| ozone-level-8hr (n: 2534, p: 73, D: 4) | Intercept | **4.69** | 4.52 (−3.6% ) | 4.385 (−6.5% ) | 4.69 (0%) |
| | V41 | **0.4146** | 0.4809 (+16% ) | 0.4343 (+4.8% ) | 0.3055 (−26.3%) |
| | V56 | **0.3559** | 0.3246 (−8.8% ) | 0.3718 (+4.5% ) | 0.351 (−1.4%) |
| | V42 | **0.2362** | 0.2628 (+11.3% ) | 0.2491 (+5.5% ) | 0.1838 (−22.2%) |
| | V13 | **0.149** | 0.1463 (−1.8% ) | 0.1651 (+10.8% ) | 0.1626 (+9.1%) |
| | V55 | **0.1438** | 0.1736 (+20.7% ) | 0.1455 (+1.2% ) | 0.1416 (−1.5%) |
| pc1 (n: 1109, p: 22, D: 3) | Intercept | **3.846** | 3.877 (+0.8% ) | 3.84 (−0.2% ) | 3.846 (0%) |
| | lOBlank | **0.544** | 0.5678 (+4.4% ) | 0.5165 (−5.1% ) | 0.5753 (+5.8%) |
| | lOBlank:lOComment | **0.3371** | 0.3327 (−1.3% ) | 0.339 (+0.6% ) | 0.2128 (−36.9%) |
| | I | **0.3226** | 0.3427 (+6.2% ) | 0.3042 (−5.7% ) | 0.3796 (+17.7%) |
| | uniq_Opnd | **0.1911** | 0.1854 (−3% ) | 0.19 (−0.6% ) | 0.1207 (−36.8%) |
| | locCodeAndComment | **0.1888** | 0.2501 (+32.5% ) | 0.1761 (−6.7% ) | 0.1853 (−1.9%) |
| pc3 (n: 1563, p: 38, D: 4) | Intercept | **3.426** | 3.742 (+9.2% ) | 3.513 (+2.5% ) | 3.426 (0%) |
| | LOC_BLANK | **0.6684** | 0.6975 (+4.4% ) | 0.6552 (−2% ) | 0.6323 (−5.4%) |
| | HALSTEAD_CONTENT | **0.4008** | 0.4438 (+10.7% ) | 0.3974 (−0.8% ) | 0.3645 (−9.1%) |
| | NUM_UNIQUE_OPERANDS | **0.1638** | 0.1682 (+2.7% ) | 0.1575 (−3.8% ) | 0.2197 (+34.1%) |
| | HALSTEAD_LEVEL | **0.1331** | 0.1263 (−5.1% ) | 0.1239 (−6.9% ) | 0.09466 (−28.9%) |
| | LOC_CODE_AND_COMMENT | **0.1328** | 0.1559 (+17.4% ) | 0.1392 (+4.8% ) | 0.1227 (−7.6%) |
| | NUMBER_OF_LINES | **0.1213** | 0.09314 (−23.2% ) | 0.1474 (+21.5% ) | 0.1031 (−15%) |
| pc4 (n: 1458, p: 38, D: 3) | Intercept | **4.275** | 3.654 (−14.5% ) | 4.38 (+2.5% ) | 4.275 (0%) |
| | LOC_CODE_AND_COMMENT | **1.507** | 1.59 (+5.5% ) | 1.471 (−2.4% ) | 1.64 (+8.8%) |
| | CYCLOMATIC_DENSITY | **0.3314** | 0.3703 (+11.7% ) | 0.3038 (−8.3% ) | 0.3219 (−2.9%) |
| | LOC_BLANK | **0.3061** | 0.3236 (+5.7% ) | 0.3011 (−1.6% ) | 0.2824 (−7.7%) |
| | CONDITION_COUNT | **0.2766** | 0.2901 (+4.9% ) | 0.2584 (−6.6% ) | 0.2782 (+0.6%) |
| pendigits (n: 10992, p: 17, D: 5) | input16 | **1.334** | 1.591 (+19.3% ) | 1.309 (−1.9% ) | 1.176 (−11.8%) |
| | input14 | **1.067** | 1.08 (+1.2% ) | 1.077 (+0.9% ) | 1.1 (+3.1%) |
| | input2 | **1.06** | 1.166 (+10% ) | 1.032 (−2.6% ) | 1.375 (+29.7%) |
| | input10 | **1.009** | 1.026 (+1.7% ) | 1.004 (−0.5% ) | 0.9265 (−8.2%) |
| | input7 | **0.7723** | 0.8284 (+7.3% ) | 0.9029 (+16.9% ) | 1.026 (+32.8%) |
| | Intercept | **-36.75** | −37.21 (+1.3% ) | −36.65 (−0.3% ) | −36.75 (0%) |
| phoneme (n: 5404, p: 6, D: 5) | Intercept | **2.866** | 2.469 (−13.9% ) | 2.834 (−1.1% ) | 2.866 (0%) |
| | V4 | **1.115** | 0.9594 (−14% ) | 1.062 (−4.8% ) | 1.271 (+14%) |
| | V1 | **0.9529** | 1.166 (+22.4% ) | 0.8947 (−6.1% ) | 0.994 (+4.3%) |

| | | | | | |
|---|---|---|---|---|---|
| | V3 | **0.789** | 0.4782 (−39.4% ) | 0.689 (−12.7% ) | 0.8698 (+10.2%) |
| | V2 | **0.7639** | 0.662 (−13.3% ) | 0.8119 (+6.3% ) | 0.6761 (−11.5%) |
| | V1:V3 | **0.7302** | 0.7381 (+1.1% ) | 0.6831 (−6.5% ) | 0.6196 (−15.1%) |
| qsar-biodeg (n: 1055, p: 42, D: 4) | Intercept | **1.872** | 2.18 (+16.5% ) | 1.829 (−2.3% ) | 1.872 (0%) |
| | V36 | **0.5753** | 0.5523 (−4% ) | 0.5769 (+0.3% ) | 0.7878 (+36.9%) |
| | V38 | **0.5006** | 0.4735 (−5.4% ) | 0.4765 (−4.8% ) | 0.493 (−1.5%) |
| | V1 | **0.2923** | 0.304 (+4% ) | 0.3126 (+6.9% ) | 0.3063 (+4.8%) |
| | V22 | **0.247** | 0.2745 (+11.1% ) | 0.2252 (−8.8% ) | 0.2285 (−7.5%) |
| | V27 | **0.2454** | 0.253 (+3.1% ) | 0.2646 (+7.8% ) | 0.2759 (+12.4%) |
| satimage (n: 6430, p: 37, D: 5) | F12attr | **1.031** | 1.161 (+12.6% ) | 1.022 (−0.9% ) | 0.3288 (−68.1%) |
| | D16attr | **0.6645** | 0.7351 (+10.6% ) | 0.6434 (−3.2% ) | 0.5567 (−16.2%) |
| | C15attr | **0.6632** | 0.5949 (−10.3% ) | 0.6421 (−3.2% ) | 0.3713 (−44%) |
| | F24attr | **0.6431** | 0.6393 (−0.6% ) | 0.6657 (+3.5% ) | 0.4283 (−33.4%) |
| | E11attr | **0.604** | 0.5513 (−8.7% ) | 0.4937 (−18.3% ) | 0.9466 (+56.7%) |
| | B8attr | **0.5692** | 0.5816 (+2.2% ) | 0.5755 (+1.1% ) | 0.5131 (−9.9%) |
| | A7attr | **0.5248** | 0.5368 (+2.3% ) | 0.4933 (−6% ) | 0.5098 (−2.9%) |
| | Intercept | **-13.61** | −13.86 (+1.8% ) | −13.51 (−0.7% ) | −13.61 (0%) |
| segment (n: 2310, p: 17, D: 3) | hue.mean | **1.577** | 1.472 (−6.7% ) | 1.573 (−0.3% ) | 1.686 (+6.9%) |
| | intensity.mean | **1.038** | 1.246 (+20% ) | 1.081 (+4.1% ) | 1.174 (+13.1%) |
| | rawgreen.mean | **0.9083** | 1.009 (+11.1% ) | 0.9338 (+2.8% ) | 0.689 (−24.1%) |
| | rawblue.mean | **0.7006** | 0.5916 (−15.6% ) | 0.6722 (−4.1% ) | 0.6068 (−13.4%) |
| | rawred.mean | **0.6916** | 0.7473 (+8.1% ) | 0.7169 (+3.7% ) | 0.3869 (−44.1%) |
| | exgreen.mean | **0.6305** | 0.7868 (+24.8% ) | 0.5637 (−10.6% ) | 0.5051 (−19.9%) |
| | Intercept | **-16.76** | −16.65 (−0.7% ) | −16.73 (−0.2% ) | −16.76 (0%) |
| semeion (n: 1593, p: 257, D: 3) | V16 | **0.5414** | 0.5967 (+10.2% ) | 0.5155 (−4.8% ) | 0.4258 (−21.4%) |
| | V229 | **0.4821** | 0.5496 (+14% ) | 0.4654 (−3.5% ) | 0.5539 (+14.9%) |
| | V77 | **0.4651** | 0.5043 (+8.4% ) | 0.4314 (−7.2% ) | 0.5042 (+8.4%) |
| | V96 | **0.4304** | 0.4823 (+12.1% ) | 0.3951 (−8.2% ) | 0.6107 (+41.9%) |
| | V15 | **0.4267** | 0.4812 (+12.8% ) | 0.4353 (+2% ) | 0.3625 (−15%) |
| | V177 | **0.417** | 0.4086 (−2% ) | 0.425 (+1.9% ) | 0.4731 (+13.5%) |
| | Intercept | **-30.14** | −29.92 (−0.7% ) | −30.24 (+0.3% ) | −30.14 (0%) |
| spambase (n: 4601, p: 58, D: 4) | Intercept | **1.827** | 1.727 (−5.5% ) | 1.397 (−23.5% ) | 1.827 (0%) |
| | word_freq_george | **0.9277** | 0.8711 (−6.1% ) | 0.9642 (+3.9% ) | 1.023 (+10.3%) |
| | word_freq_hp | **0.8724** | 0.8814 (+1% ) | 0.8138 (−6.7% ) | 0.9547 (+9.4%) |

| | | | | |
|---|---|---|---|---|
| | char_freq_.21 | **0.4842** | 0.4836 (−0.1% ) | 0.5165 (+6.7% ) | 0.858 (+77.2%) |
| | capital_run_length_longest | **0.4376** | 0.4276 (−2.3% ) | 0.426 (−2.7% ) | 0.4318 (−1.3%) |
| | capital_run_length_total | **0.4237** | 0.4077 (−3.8% ) | 0.4321 (+2% ) | 0.4258 (+0.5%) |
| | char_freq_.24 | **0.372** | 0.3631 (−2.4% ) | 0.3414 (−8.2% ) | 0.6447 (+73.3%) |
| splice (n: 3190, p: 61, D: 5) | attribute_32.G | **0.7944** | 0.771 (−2.9% ) | 0.8258 (+4% ) | 0.7655 (−3.6%) |
| | attribute_31.A | **0.7598** | 0.7031 (−7.5% ) | 0.7579 (−0.3% ) | 0.7452 (−1.9%) |
| | attribute_30.C | **0.7277** | 0.6369 (−12.5% ) | 0.798 (+9.7% ) | 0.722 (−0.8%) |
| | attribute_30.T | **0.6651** | 0.811 (+21.9% ) | 0.6216 (−6.5% ) | 0.6703 (+0.8%) |
| | attribute_32.C | **0.6098** | 0.6685 (+9.6% ) | 0.4642 (−23.9% ) | 0.6048 (−0.8%) |
| | attribute_32.A | **0.6011** | 0.4873 (−18.9% ) | 0.6426 (+6.9% ) | 0.6169 (+2.6%) |
| | Intercept | **-3.597** | −4.027 (+12% ) | −3.588 (−0.3% ) | −3.597 (0%) |
| steel-plates-fault (n: 1941, p: 28, D: 5) | V14 | **0.8236** | 0.816 (−0.9% ) | 0.7835 (−4.9% ) | 1.181 (+43.4%) |
| | V25 | **0.8116** | 0.6918 (−14.8% ) | 0.9687 (+19.4% ) | 0.8836 (+8.9%) |
| | V11 | **0.7111** | 0.8872 (+24.8% ) | 0.7183 (+1% ) | 0.7845 (+10.3%) |
| | V12 | **0.6505** | 0.6886 (+5.9% ) | 0.7197 (+10.6% ) | 0.7353 (+13%) |
| | Intercept | **-13.57** | −13.88 (+2.3% ) | −13.3 (−2% ) | −13.57 (0%) |
| texture (n: 5500, p: 41, D: 3) | V23 | **2.455** | 2.444 (−0.4% ) | 2.389 (−2.7% ) | 2.505 (+2%) |
| | V10 | **2.006** | 2.037 (+1.5% ) | 1.87 (−6.8% ) | 1.206 (−39.9%) |
| | V3 | **1.91** | 2.004 (+4.9% ) | 1.893 (−0.9% ) | 1.209 (−36.7%) |
| | V30 | **1.63** | 1.515 (−7.1% ) | 1.558 (−4.4% ) | 1.083 (−33.6%) |
| | V6 | **0.8912** | 0.8164 (−8.4% ) | 0.8066 (−9.5% ) | 1.208 (+35.5%) |
| | Intercept | **-39.86** | −41.69 (+4.6% ) | −40.01 (+0.4% ) | −39.86 (0%) |
| tic-tac-toe (n: 958, p: 10, D: 3) | middle.middle.square.o | **2.332** | 2.278 (−2.3% ) | 2.198 (−5.7% ) | 2.012 (−13.7%) |
| | middle.middle.square.x | **1.882** | 1.981 (+5.3% ) | 1.764 (−6.3% ) | 1.554 (−17.4%) |
| | top.left.square.o | **1.71** | 1.696 (−0.8% ) | 1.535 (−10.2% ) | 1.501 (−12.2%) |
| | bottom.right.square.o | **1.703** | 1.827 (+7.3% ) | 1.872 (+9.9% ) | 1.438 (−15.6%) |
| | top.right.square.x | **1.506** | 1.6 (+6.2% ) | 1.643 (+9.1% ) | 1.349 (−10.4%) |
| | Intercept | **-1.818** | −2.839 (+56.2% ) | −1.341 (−26.2% ) | −1.818 (0%) |
| vehicle (n: 846, p: 19, D: 2) | ELONGATEDNESS | **1.544** | 1.468 (−4.9% ) | 1.596 (+3.4% ) | 1.326 (−14.1%) |
| | DISTANCE_CIRCULARITY | **1.168** | 1.185 (+1.5% ) | 1.209 (+3.5% ) | 0.9147 (−21.7%) |
| | MAX.LENGTH_RECTANGULARITY | **1.02** | 1.063 (+4.2% ) | 0.9321 (−8.6% ) | 1.032 (+1.2%) |
| | KURTOSIS_ABOUT_MINOR | **0.6494** | 0.7477 (+15.1% ) | 0.7149 (+10.1% ) | 0.3895 (−40%) |
| | SKEWNESS_ABOUT_MAJOR | **0.6262** | 0.5876 (−6.2% ) | 0.6585 (+5.2% ) | 0.6867 (+9.7%) |
| | Intercept | **-5.312** | −5.066 (−4.6% ) | −5.296 (−0.3% ) | −5.312 (0%) |

| | | | | | |
|---|---|---|---|---|---|
| vowel (n: 990, p: 13, D: 3) | Feature_0 | **2.975** | 2.916 (−2% ) | 3.096 (+4.1% ) | 2.701 (−9.2%) |
| | Feature_4 | **2.426** | 2.455 (+1.2% ) | 2.449 (+0.9% ) | 2.429 (+0.1%) |
| | Feature_1 | **2.368** | 2.45 (+3.5% ) | 2.433 (+2.7% ) | 2.592 (+9.5%) |
| | Feature_0:Feature_1 | **1.517** | 1.476 (−2.7% ) | 1.392 (−8.2% ) | 1.273 (−16.1%) |
| | Intercept | **-39.77** | −38.88 (−2.2% ) | −40.37 (+1.5% ) | −39.77 (0%) |
| wall-robot-navigation (n: 5456, p: 25, D: 3) | V15 | **2.281** | 2.427 (+6.4% ) | 2.339 (+2.5% ) | 3.076 (+34.9%) |
| | V14:V15 | **1.624** | 1.621 (−0.2% ) | 1.777 (+9.4% ) | 1.087 (−33.1%) |
| | V14 | **1.381** | 1.704 (+23.4% ) | 1.53 (+10.8% ) | 1.387 (+0.4%) |
| | V20 | **1.246** | 1.238 (−0.6% ) | 1.375 (+10.4% ) | 1.367 (+9.7%) |
| | V19 | **0.8858** | 0.8706 (−1.7% ) | 0.9571 (+8% ) | 1.485 (+67.6%) |
| | Intercept | **-10.77** | −11.56 (+7.3% ) | −10.79 (+0.2% ) | −10.77 (0%) |
| wdbc (n: 569, p: 31, D: 5) | Intercept | **2.367** | 3.815 (+61.2% ) | 3.279 (+38.5% ) | 2.367 (0%) |
| | V28 | **1.124** | 1.109 (−1.3% ) | 1.1 (−2.1% ) | 1.502 (+33.6%) |
| | V21 | **0.9865** | 0.9992 (+1.3% ) | 0.9409 (−4.6% ) | 1.469 (+48.9%) |
| | V8 | **0.9086** | 0.8257 (−9.1% ) | 0.8677 (−4.5% ) | 1.189 (+30.9%) |
| | V22 | **0.7937** | 0.8219 (+3.6% ) | 0.7633 (−3.8% ) | 0.8532 (+7.5%) |
| | V23 | **0.7138** | 0.6631 (−7.1% ) | 0.6834 (−4.3% ) | 0.9698 (+35.9%) |
| wilt (n: 4839, p: 6, D: 3) | Intercept | **6.194** | 6.613 (+6.8% ) | 6.117 (−1.2% ) | 6.194 (0%) |
| | Mean_G:Mean_R | **3.12** | 3.096 (−0.8% ) | 3.103 (−0.5% ) | 1.99 (−36.2%) |
| | Mean_G | **2.903** | 3.066 (+5.6% ) | 2.981 (+2.7% ) | 2.175 (−25.1%) |
| | Mean_R | **2.829** | 2.775 (−1.9% ) | 2.851 (+0.8% ) | 1.73 (−38.8%) |
| | SD_Plan | **0.3629** | 0.3611 (−0.5% ) | 0.3538 (−2.5% ) | 0.3564 (−1.8%) |
| | Mean_NIR | **0.3565** | 0.3651 (+2.4% ) | 0.3686 (+3.4% ) | 0.3257 (−8.6%) |

