# OpenReview forum: "Fast Estimation of Partial Dependence Functions using Trees"
_ICML.cc/2025/Conference — ICML 2025 poster_

### Official Review · Reviewer_P6Hy · 2025-03-11

**Overall Recommendation:** 2

**Summary:**

This paper proposes an efficient tree-based algorithm called FastPD for estimating partial dependence (PD) functions and extracting functional decompositions. The authors highlight how it can unify PD plots, Shapley values, and higher-order interaction effects under one consistent framework. By carefully caching and reusing computations, the method achieves gains over naive or path-dependent approaches, particularly when features are correlated. The main thrust is that FastPD provides both improved consistency and reduced computational complexity, compared to commonly used baselines like path-dependent TreeSHAP. In practice, it claims a user to quickly derive multiple interpretability artifacts from a single model-agnostic foundation.

**Claims And Evidence:**

The authors study the inconsistency of TreeSHAP, more specifically when features are correlated.

**Essential References Not Discussed:**

References are fine, but a recent study is missed:
Unifying Feature-Based Explanations with Functional ANOVA and Cooperative Game Theory


[1] Hiabu, M., Meyer, J. T., and Wright, M. N. Unifying
local and global model explanations by functional decomposition of low dimensional structures

**Experimental Designs Or Analyses:**

While the authors include simulation studies—showing that the method works better on synthetic data compared to TreeSHAP—they do not examine any real-world datasets or large-scale industrial applications. This can make it difficult to assess how well FastPD copes with real-world issues such as messy data distributions, large numbers of features with intricate dependencies, and numeric stability in high-dimensional spaces. This is particularly important because the proposed method is more an algorithmic innovation.

**Methods And Evaluation Criteria:**

The evaluation is quite weak; FastPD is only compared with two versions of TreeSHAP in one experiment of adding more background samples over a relatively small XGBoost. The evaluation is merely over the synthetic data sets; We cannot draw any conclusion from the experimetns provided in the article.

**Other Comments Or Suggestions:**

The title could be more informative by mentioning that FastPD is particularly for tree-based machine learning models.

**Other Strengths And Weaknesses:**

I think the most crucial problem is the experimental setup; the evaluative measures are not rigorous, and the experiments are only conducted on small, limited synthetic data sets. TreeSHAP is also the only benchmarks. More comprehensive experiments are required, where FastPD is compared over real data sets according to some meaningful evaluative measures.
This is particularly important for FastDP because it puts forward more an algorithmic way of computing PD than a rigorous theoretical one.

**Questions For Authors:**

The author should discuss that not all the functional components could be acquired as there are exponentially many such components.

**Relation To Broader Scientific Literature:**

There are several studies at the intersection of functional decomposition and PDP. FastPD could be a worthy added value to the literature, but it definitely requires far more experiments and probably more theoretical studies on the behavior of the method.

**Theoretical Claims:**

The paper studies the consistency of the proposed method and the inconsistency of TreeSHAP.

---

> ### Author Rebuttal · Authors · 2025-04-01
>
> Thank you for your comments.
>
> In our response to the first reviewer we have conducted further experiments of our method on real data (see **1. Real-world application** in our response to R1). In our response to the second reviewer, we advice using shallow trees to reduce complexity for large-scale datasets with many features (see **2. High-complexity models** in our response to R2). We believe that our method is still very applicable even in high-dimensions, provided that the model is not overly complex. We have not encountered numerical instability in our experiments with real data; since with many samples, the estimation of partial dependence (PD) functions becomes more reliable, not less. Moreover, practitioners are typically interested in marginal effects and low-order interactions, because they are easier to interpret. In all cases, the actual quantity being computed is a just an empirical average, for which `FastPD` can provide a speed-up when the model is a tree-based model.
>
> We agree that not all functional components could or should be acquired, since for higher-order interactions the effects of these components are usually very close to zero, we will make this point more clear in the paper. This further motivates the use of shallow trees, as they limit the fitting of higher-order interactions and preserve interpretability.
> As a remark, in the package implementation of our method, we provide the user the option to specify the order of the effect they wish to acquire.
>
> For the reference that we had inadvertently missed, perhaps it was the following you meant?
>
> > Fumagalli, F., Muschalik, M., Hüllermeier, E., Hammer, B., & Herbinger, J. (2024). *Unifying Feature-Based Explanations with Functional ANOVA and Cooperative Game Theory*. arXiv preprint arXiv:2412.17152.
>
> We have already cited Hiabu et al. multiple times, but were not yet aware of Fumagalli et al. and will ensure it is cited in the revised version.
>
> Finally, we agree that the paper title could be clearer. A possible revision would be:
> **"Fast Estimation of Partial Dependence Functions using Trees"**,
> which more accurately reflects the tree-based nature of our approach.

---

### Official Review · Reviewer_XQ8h · 2025-03-13

**Overall Recommendation:** 3

**Summary:**

This paper proposes FastPD, an efficient, and consistent algorithm for estimating Partial Dependence (PD) functions in tree-based models. FastPD addresses computational inefficiencies present in existing methods by decomposing the estimation process into a two-step procedure: a tree augmentation step leveraging background data and an evaluation step for calculating PD functions. The authors show that FastPD improves computational complexity from quadratic to linear with respect to the number of samples. Furthermore, the paper demonstrates that existing methods, like TreeSHAP-path, are inconsistent when features are correlated. Experimental results indicate that FastPD provides more accurate estimates of PD-based interpretations, including SHAP values, compared to competing algorithms.

**Claims And Evidence:**

The claims regarding the computational efficiency and consistency of FastPD are convincingly supported through theoretical propositions and experimental validations. However, the claim about significant differences between FastPD and TreeSHAP-path estimates is primarily demonstrated through simulation experiments with moderate feature correlation. Additional real-world experiments could further strengthen these claims.

**Essential References Not Discussed:**

The paper is comprehensive in its citations; however, additional references exploring the scalability of SHAP (beyond tree-based methods) would enhance context for readers interested in broader applicability.

- Jethani, Neil, et al. "Fastshap: Real-time shapley value estimation." International conference on learning representations. 2021.
- Wang, Guanchu, et al. "Accelerating shapley explanation via contributive cooperator selection." International Conference on Machine Learning. PMLR, 2022.

**Experimental Designs Or Analyses:**

The experimental design and analysis presented in the paper appear sound. The experiments include comparisons of computational runtime, estimation error (mean squared error), and consistency across different correlation settings. No immediate issues were found in the methodology or its implementation.

**Methods And Evaluation Criteria:**

The methods and evaluation criteria used in the paper are appropriate for the stated problem of estimating PD functions efficiently. The paper clearly outlines how the complexity and consistency of the algorithms are assessed. Using simulations with varying levels of feature correlation and benchmarking against established methods (TreeSHAP-path, TreeSHAP-int, VanillaPD) is sensible and effective.

**Other Comments Or Suggestions:**

The paper could be clearer in distinguishing the practical implications of model-based PD functions versus ground-truth PD functions in realistic scenarios.

Minor typo found: "adddition" should be "addition" (Section 1.1).

**Other Strengths And Weaknesses:**

Strengths include clear contributions, detailed and rigorous mathematical arguments regarding algorithmic complexity, and practical relevance for model interpretability in correlated feature settings. The experimental validation is thorough and clearly presented.

A weakness is that the scalability discussion focuses primarily on moderately deep trees and relatively simple experimental settings. The impact of scaling to larger, real-world datasets and more complex tree-based ensembles (deeper or broader models) is less clear.

**Questions For Authors:**

How does FastPD scale when applied to larger and deeper tree-based ensembles (e.g., random forests or deep gradient boosting trees)? Clarifying this could strengthen practical applicability.

Could you provide guidance or recommendations on choosing the optimal number of background samples for FastPD in different practical scenarios?

**Relation To Broader Scientific Literature:**

The paper positions itself clearly within the broader scientific literature, notably referencing foundational work on SHAP values, PD functions, and functional decomposition. It builds directly on TreeSHAP-related methods and addresses previously identified limitations (such as inconsistency due to feature correlation).

**Theoretical Claims:**

The theoretical claim regarding the inconsistency of TreeSHAP-path is clearly stated, and its correctness is supported by a formal proof provided in the supplementary material. I reviewed the proof in the appendix and found it sound and convincing.

---

> ### Author Rebuttal · Authors · 2025-04-01
>
> Thank you for your comments, we address them below.
>
> 1. **Real-world applications:**
>    We have addressed this in our reply to the first reviewer. Please see **1. Real-world application** in our response to R1.
>
> 2. **High-complexity models:**
>    The computational complexity of both our algorithm and the baselines scales exponentially with depth, which motivates the use of smaller depth values. Additionally, increasing depth captures higher-order interactions that are inherently less interpretable. Gradient Boosting methods such as XGBoost fit shallow trees well; we therefore recommend using a depth of 5 or less in practice. We will clarify this point in the paper.
>
> 3. **Referencing SHAP scalability:**
>    We will add some references for non-tree-based SHAP algorithms in our introduction.
>
> 4. **Distinguishing model PD and ground-truth PD:**
>    We will add the following to the application section:
>    *"We see that FastPD is well-suited for estimating the model PD, though it may deviate from the underlying ground truth PD. In practical settings where accurately capturing the relationship between the predictors and response is crucial, we recommend using FastPD as an initial visualization tool, which is to be complemented by other statistical methods for estimating continuous treatment effects, see e.g. Kennedy, E. H., Ma, Z., McHugh, M. D., & Small, D. S. (2017). Non-parametric methods for doubly robust estimation of continuous treatment effects. Journal of the Royal Statistical Society Series B: Statistical Methodology, 79(4), 1229-1245.
> and Chernozhukov, V., Chetverikov, D., Demirer, M., Duflo, E., Hansen, C., Newey, W., & Robins, J. (2018). Double/debiased machine learning for treatment and structural parameters. The Econometrics Journal, 21(1),  C1–C68.."*
>
> 5. **Choice of background sample size:**
>    We now note in the main text that in practice, one should use all available data as background. If this is not feasible, a smaller background sample size can be chosen (e.g., $n_b = 100$) and compared to a slightly larger one (e.g., $n_b = 150$). If the resulting PD functions are similar, then the smaller sample size is likely sufficient.
>
>    As a remark, in our experiments we have observed that augmenting the tree is typically fast, so the computational cost of using more background samples is low. The majority of computation time is instead spent on evaluating the explanation points.

---

### Official Review · Reviewer_KVb7 · 2025-03-14

**Overall Recommendation:** 4

**Summary:**

This paper introduces FastPD, a novel tree-based algorithm for estimating partial dependence (PD) functions, which are central for interpreting machine learning models via SHAP values. The paper identifies a critical limitation in the commonly used TreeSHAP-path method: while TreeSHAP-path is theoretically exact under feature independence, its reliance on conditioning on the decision path can lead to biased SHAP estimates when features are correlated. FastPD overcomes this issue by decoupling the computation into an augmentation phase—where the full background dataset is used to precompute empirical probabilities—and an efficient evaluation phase that reuses these values to compute the PD functions exactly. The result is a method that is both computationally efficient and robust to feature correlations.

**Claims And Evidence:**

Overall, the claims are well supported by clear theoretical derivations and convincing simulation experiments, though real-world dataset experiments would further strengthen the evidence.

**Essential References Not Discussed:**

N/A.

**Experimental Designs Or Analyses:**

The experiments are based on simulated data with controlled levels of feature correlation using tree-based models (e.g., XGBoost). The comparisons include multiple methods (VanillaPD, TreeSHAP-path, TreeSHAP-int, and FastPD) and evaluate both accuracy (MSE) and computational runtime. The experimental analyses are sound and clearly demonstrate that FastPD achieves lower estimation error and faster computation. While the designs are rigorous for simulation studies, incorporating additional experiments on real-world datasets would further validate the approach in practical settings.

**Methods And Evaluation Criteria:**

The proposed method is specifically designed for tree-based models, leveraging the structure to precompute necessary statistics (augmentation) and then efficiently evaluate PD functions. This two-step approach is both novel and well-motivated.
Overall, the methods and evaluation criteria make sense for the problem of model interpretability, although additional experiments on real-world datasets would enhance the evaluation.

**Other Comments Or Suggestions:**

Consider including experiments on real-world datasets to validate the method’s practical relevance. For example, what can practitioners expect to see in practice? Is the bias of TreeSHAP really that bad? Can it lead to "catastrophic" situations where you attribute significantly less importance to something with with TreeSHAP vs FastPD (we now know it is consistent, but can you make a case for why this is so bad in practice in a relatively varied collection of benchmark datasets?). I think these extensions will really increase the impact of the paper, which is why I want to be transparent and tell the authors that I will definitely raise my score from a 3, if a stronger set of experiments with interpretations on the differences between FastPD and other methods is presented! I think it will really convince readers to use this method!


Additional illustrative diagrams of the augmentation and evaluation steps in FastPD would improve clarity.

**Other Strengths And Weaknesses:**

Strengths:
Novel decoupling of augmentation and evaluation steps to compute exact PD functions.
Clear mathematical exposition linking SHAP values, partial dependence functions, and the issues with conditioning in TreeSHAP-path.
Comprehensive simulation experiments that highlight both estimation accuracy and runtime improvements.

Weakness:
Lack of evaluation on real-world datasets limits the demonstration of practical applicability.
The algorithm’s presentation could benefit from more intuitive diagrams or flowcharts to aid understanding.
Reproducibility details (e.g., explicit hyperparameter settings) could be more thoroughly described in the main text.

**Questions For Authors:**

Code seems a little messy, do the authors intend to release an easy-to-use package? I think it would be helpful for people to adopt this approach.

**Relation To Broader Scientific Literature:**

In general this is a good contribution to the literature on interpretability, and shapley values, particularly addressing the bias with existing approaches like TreeSHAP.

**Theoretical Claims:**

The theoretical arguments are well-presented and, at a high level, correct. Minor technical details may need further scrutiny, but no major issues were identified.

---

> ### Author Rebuttal · Authors · 2025-04-01
>
> Thank you for your comments, we address them below.
>
> 1. **Real-world application:**
>
>    - **Benchmark:**
>      For an additional comparison between `FastPD`, `FastPD-100`, `FastPD-50`, and the path-dependent method, we have now added a benchmark considering 33 regression and 29 classification datasets from the OpenML-CTR23 Task Collection and the OpenML-CC18 Task Collection. For a set $S \subseteq \{1,\dots,d\}$, we compute the importance measure of the estimated functional component $\hat{m}_S$ by $\hat{E}[|m_S(x)|]$, where $\hat{E}[\cdot]$ denotes the empirical mean.
>      For each dataset, we record the importance of the components $\hat{m}_S$ that are ranked in the top 5 by any of the four methods (`FastPD`, `FastPD-100`, `FastPD-50`, path-dependent).
>      The results are available here:
>      [https://www.dropbox.com/scl/fi/ujdikzu8iyrn53ha03qc4/summary.pdf?rlkey=ypnkbdhjwgylr38ylj16ugjzj&st=f97bol4d&dl=0](https://www.dropbox.com/scl/fi/ujdikzu8iyrn53ha03qc4/summary.pdf?rlkey=ypnkbdhjwgylr38ylj16ugjzj&st=f97bol4d&dl=0)
>      We observe that while results are consistent across methods for some datasets, substantial differences can emerge in others. In several cases the differences are >100%, i.e., the feature importance values from the path-dependent method are more than twice as high as those from FastPD.
>
>    - **Concrete example:**
>      We highlight one dataset (`adult`) where `FastPD` and the path-dependent method gave qualitatively different insights. The following plot based on a single model obtained from one run of hyperparameter search. We have plotted the interaction effect between age and relationship status:
>      $m_{\text{age,relationship}}$.
>      [https://www.dropbox.com/scl/fi/8wx0xj34rglhst6tm8eb0/adult.pdf?rlkey=nrgz74xjrd1udoxgrhohwx6au&st=yclm7q6o&dl=0](https://www.dropbox.com/scl/fi/8wx0xj34rglhst6tm8eb0/adult.pdf?rlkey=nrgz74xjrd1udoxgrhohwx6au&st=yclm7q6o&dl=0)
>      The path-dependent method indicates no significant interaction effect during working age (35–60), suggesting that age affects husbands and wives similarly in this range. Outside this interval, it estimates a more positive effect for wives than for husbands.
>      Using `FastPD`, we get a different picture: Between ages 30 and 65, the effect is more positive for husbands than for wives, with the advantage reversing outside this age range.
>
> 2. **Flowcharts/diagrams:**
>    We agree that a flowchart and illustration would help in explaining the algorithm, and will add it for the camera-ready version.
>
> 3. **Reproducibility details:**
>    We will modify the beginning of the experiments section to include an overview of our hyperparameter settings.
>
> 4. **Package implementation:**
>    A package implementation exists, including automated plot functions. However, due to anonymity, we prefer not to share it at this point, but it will be made available in the camera-ready version.

---

> > ### Comment · Reviewer_KVb7 · 2025-04-05
> >
> > These are very interesting findings, which I believe significantly strengthen the story behind this paper! I have raised my score to a 4 and I am happy to recommend acceptance!

---

### Decision · Program_Chairs · 2025-05-01

**Decision:**

Accept (poster)

**Comment:**

The paper introduces FastPD, a novel algorithm for estimating partial dependence (PD) functions for tree-based models. These functions are crucial in computing, for instance, SHAP explanations or partial dependence plots. FastPD has a low complexity and, unlike other fast algorithm such as path-dependent TreeSHAP, it is exact in the case of feature correlation.

The strengths of the paper have been listed by the reviewers as the clear motivation, novelty, and soundness of the proposed method, the clear mathematical derivation, the link drawn in the paper between SHAP and PD functions, and the good empirical performance in terms of estimation quality and computing times

All reviewers expressed concerns about the lack of experiments on real-world datasets. In their response, the authors have carried out new experiments on a large set of real-world benchmarks to demonstrate the difference between FastPD and path-dependent TreeSHAP, as suggested by reviewer KVb7. They also identified a dataset where FastPD and TreeSHAP lead to different conclusions. While reviewer P6Hy still believes more experiments are necessary, reviewer KVb7 has increased their score.

Although the reviewers don’t unanimously recommend acceptance, I believe incorporating the new experiments will strengthen the paper that already presents a solid algorithmic and theoretical contribution and is well-presented. Therefore, I recommend acceptance. I trust the authors to incorporate the new results into the paper and make all other changes promised in their responses. I agree with reviewer P6Hy that the title of the paper should explicitly mention that the approach is focused on tree-based models.